# Xylose and shimate transporters facilitates microbial consortium as a chassis for benzylisoquinoline alkaloid production

Meirong Gao[1,2,7], Yuxin Zhao[1,2,7], Zhanyi Yao [1,2], Qianhe Su[1], Payton Van Beek[1] & Zengyi Shao [1,2,3,4,5,6] ✉

Plant-sourced aromatic amino acid (AAA) derivatives are a vast group of compounds with broad applications. Here, we present the development of a yeast consortium for efficient production of (S)-norcoclaurine, the key precursor for benzylisoquinoline alkaloid biosynthesis. A xylose transporter enables the concurrent mixed-sugar utilization in *Scheffersomyces stipitis*, which plays a crucial role in enhancing the flux entering the highly regulated shikimate pathway located upstream of AAA biosynthesis. Two quinate permeases isolated from *Aspergillus niger* facilitates shikimate translocation to the co-cultured *Saccharomyces cerevisiae* that converts shikimate to (S)-norcoclaurine, resulting in the maximal titer (11.5 mg/L), nearly 110-fold higher than the titer reported for an *S. cerevisiae* monoculture. Our findings magnify the potential of microbial consortium platforms for the economical de novo synthesis of complex compounds, where pathway modularization and compartmentalization in distinct specialty strains enable effective fine-tuning of long biosynthetic pathways and diminish intermediate buildup, thereby leading to increases in production.

Plant-sourced aromatic amino acid (AAA) derivatives are a highly diverse group of compounds that are widely used as food additives, nutraceuticals, pharmaceuticals, and building blocks of drugs with considerable commercial values[1]. However, their biosynthesis in native hosts is tightly regulated and tissue specific. The products are mixed with many structurally similar intermediates, leading to low yields and costly extraction processes[2]. Some plants producing highly desired AAA derivatives (i.e., *Saussurea involucrata* producing hispidulin, rutin, and syringin) are barely cultivable or endangered[3,4]. Chemical synthesis of AAA derivatives is markedly retarded by their complex structures, imposing stringent requirements for specific stereochemistry[5]. Accordingly, the development of synthetic microbial platforms as surrogates has garnered increased attention. Most studies employ model microorganisms, such as *Escherichia coli* and *Saccharomyces*

*cerevisiae*, with special emphasis on *S. cerevisiae* due to its amenability to functional expression of endoplasmic reticulum (ER)-associated cytochrome P450s[6] and complex post-translational modifications[7].

Koopman et al. reported de novo production of the exemplary flavonoid, naringenin, from glucose with a titer of 109 mg/l[8]. Further extension and modification of the naringenin pathway led to the synthesis of additional flavonoid compounds, such as kaempferol, quercetin[9], phloretin, nothofagin, and trilobatin[10], with titers ranging from 20 to 60 mg/l. The biosynthesis pathway of resveratrol, a natural polyphenol from the stilbenoid family, was also reconstituted in *S. cerevisiae*[11]. The combination of a stepwise metabolic engineering strategy with a two-phase fed-batch fermentation enabled a final titer of 531.41 mg/l. In the benzylisoquinoline alkaloid (BIA) branch, the complete biosynthesis of thebaine, hydrocodone[12], and noscapine[13]

[1]Department of Chemical and Biological Engineering, Iowa State University, Ames, IA, USA. [2]NSF Engineering Research Center for Biorenewable Chemicals, Iowa State University, Ames, IA, USA. [3]Interdepartmental Microbiology Program, Iowa State University, Ames, IA, USA. [4]Bioeconomy Institute, Iowa State University, Ames, IA, USA. [5]The Ames Laboratory, Ames, IA, USA. [6]DOE Center for Advanced Bioenergy and Bioproducts Innovation, University of Illinois at Urbana-Champaign, Urbana, IL, USA. [7]These authors contributed equally: Meirong Gao, Yuxin Zhao. ✉e-mail: zyshao@iastate.edu

from glucose via the coordinated expression of 20–30 enzymes from various species in *S. cerevisiae* represents important milestones in building complex microbial synthetic platforms for the synthesis of plant-sourced natural products, albeit the production levels are only a few mg/l (noscapine) or μg/l (thebaine and hydrocodone).

The inefficient production is mainly caused by the distinct challenges present in each module of the long pathways. The upstream pathway is native to the microbial hosts and is directly linked to their primary metabolism; therefore, hosts employ hierarchically sophisticated controls to minimize unnecessary energy expenditure for AAA synthesis[14]. For the downstream foreign add-on steps, it is challenging to functionally express combinations of genes from plants, mammals, bacteria, and fungi, as well as simultaneously balance their expression levels (among themselves and relative to the upstream enzymes). Impairments in the pathway could result in intermediate buildups and thus, significantly decrease the yield of final product. In the past two decades for BIA biosynthesis, extensive efforts have focused on: (1) reducing enzyme promiscuity (e.g., the dopamine pathway)[15], (2) creating enzyme chimeras to avoid incorrect processing and glycosylation occurring in the wrong compartment (e.g., the thebaine pathway)[12], (3) reinforcing cofactor regeneration to improve the activities of cytochrome P450s and AAA hydroxylase (e.g., tetrahydrobiopterin in the melatonin and thebaine pathways)[12,16,17], (4) pairing enzymes from multiple species (e.g., the sanguinarine pathway)[18], (5) titrating gene copy number and spatially separating certain enzymes to intervene in side reactions (e.g., the morphine pathway)[19], and (6) optimizing enzyme activity (especially P450s) by testing promoters of different strengths and regulation patterns (e.g., the sanguinarine pathway)[18].

In the post-CRISPR era, rapid advancement in the exploration of unique microbial species challenges the status of *E. coli* and *S. cerevisiae* as classical microbial cell factories[20]. Many nonconventional microorganisms are of great interest in industrial applications (e.g., high temperature and low pH) due to their innate traits with respect to their unique desirable physiology, metabolism, biosynthesis, and fermentation capacity, which are acquired through a long-term natural evolution in specific environments[21,22]. For the upstream module of the AAA biosynthetic pathways, our previous study revealed a sevenfold higher shikimate titer in *Scheffersomyces stipitis* than *S. cerevisiae*[23], as the active pentose phosphate pathway (PPP) in *S. stipitis* renders a higher availability of erythrose-4-phosphate (E4P), the rate-limiting precursor identified in *S. cerevisiae*[24]. The preliminary success in optimizing the upstream module suggests that *S. stipitis* could serve as a potentially better host than *S. cerevisiae* to produce AAA derivatives. However, the current shortage of readily usable multi-auxotrophic strains, high-copy plasmids, and multiplex genome editing protocols for *S. stipitis* hampers its application in AAA biosynthesis. Moreover, the daunting issues, such as the expression of challenging enzymes, ER stress, unrevealed competing pathways, and cofactor scarcity, as encountered in *S. cerevisiae*, will most likely appear in this host. Recent advances in building synthetic microbial consortia have manifested their superiority in metabolic burden reduction, pathway compartmentalization and balancing, and host specialization[25–28].

In this study, to resolve issues specific to individual pathway segmentations, we build a yeast consortium to leverage the highly active upstream module provided by *S. stipitis* and the extensively optimized downstream modules in *S. cerevisiae* (Fig. 1). Considering that xylose assimilation can further enhance in the carbon flux of PPP, a xylose transporter encoded by *Spathaspora passalidarum XUT1* is identified and characterized. We prove that its overexpression is crucial to enabling simultaneous mixed-sugar conversion and consequently ameliorating the upstream flux to shikimate production. However, as a native metabolite, shikimate shows a unidirectional translocation, i.e., not assimilated by *S. cerevisiae*. This is enabled by the discovery of two quinate permeases

that show high capacity for shikimate transport. Independent optimization of two modules, efficient transfer of the connecting molecule, and an easily adjustable ratio between the two species enable production of (*S*)-norcoclaurine at 11.5 mg/l, nearly 110-fold higher than the titer reported for an *S. cerevisiae* monoculture[15]. Altogether, this study highlights the advantages of using consortia to reconstitute long and complex metabolic pathways and accelerate the development of microbial platforms for economical production of complex plant-specialized natural products.

## Results
### Refactored xylose utilization pathway ameliorated the upstream flux

Compartmentalization enabled by a consortium allows us to address issues specific to each individual module of a complex pathway. Previously, we constructed a shikimate-producing *S. stipitis* strain, which, prior to extensive optimization, outperformed *S. cerevisiae* regarding the flux entering the upstream module of AAA biosynthesis[23]. Considering adequate evidence suggests downstream conversion efficiency constrains BIA production[12,13,29], we hypothesized that a highly optimized upstream module in *S. stipitis* would save resources in the co-culture for the enrichment of the *S. cerevisiae* population carrying the downstream module.

In the current collection of yeast species, *S. stipitis* is attractive because of its superior native xylose-fermenting capacity[30]. As xylose is assimilated into the central metabolism through PPP, improved simultaneous utilization of glucose and xylose would presumably increase the E4P supply, leading to a higher flux toward the upstream module of the AAA biosynthesis pathway (e.g., the segment for shikimate synthesis). However, persistent carbon catabolite repression (CCR) severely prohibits *S. stipitis* from efficiently assimilating xylose in the presence of glucose, a ubiquitous cellular phenomenon observed in bacteria, yeasts, and other fungi[31,32]. It is worth clarifying that for synthesizing compounds derived from the downstream module of the AAA pathway, incorporation of the xylose-utilization module is not based on the potential of using xylose-containing biomass as a low-cost feedstock. The xylose pathway is added simply because of its positive influence on the landscape of carbon flux distribution that could potentially benefit the AAA upstream module.

To elucidate the cause of CCR in *S. stipitis*, we performed RNA-seq analysis of the strains grown in cultures containing the following carbon sources: (1) pure glucose, (2) pure xylose, (3) mixture of xylose and glucose (Supplementary Fig. 1). All three genes in the xylose-utilizing pathway (i.e., *XYL1*, *XYL2*, and *XYL3*) and two genes involved in PPP (i.e., *TKT1* and *TAL1*) were identified to be substantially downregulated in both the glucose condition (e.g., by 3.8–56.3 fold at 15 h) and the mixed-sugar condition (e.g., by 4.0–21.3 fold at 15 h) (Supplementary Table 1). To overcome glucose repression at the transcriptional level, we constructed a plasmid, designated as pMG-xyl (Supplementary Data 1 and Supplementary Fig. 2), by replacing the native promoters of these five genes with strong constitutive promoters previously isolated from *S. stipitis*, whose activity is proven insensitive to carbon sources[23]. The plasmid, pMG-xyl, was transformed into the *S. stipitis* FPL-UC7 strain[33], yielding the strain, Ss-xyl. High cell-density fermentation (HCDF, with an initial $OD_{600nm}$ at ~10) was conducted to test the sugar usage of the strain Ss-xyl. In contrast to the strict CCR displayed by the control strain (named Ss-WT xyl carrying the wild-type xylose pathway) that consumed 1.5 g/l xylose, the strain Ss-xyl metabolized 4.4 g/l xylose in the mixed sugars during a 5-day fermentation (Fig. 2a). This result indicated that CCR in *S. stipitis* was partially attributable to repressed transcription of the genes involved in intracellular xylose assimilation. Other bottlenecks, such as efficient xylose import and global regulation[32], exist and need to be addressed to enable a more substantial mitigation of CCR.

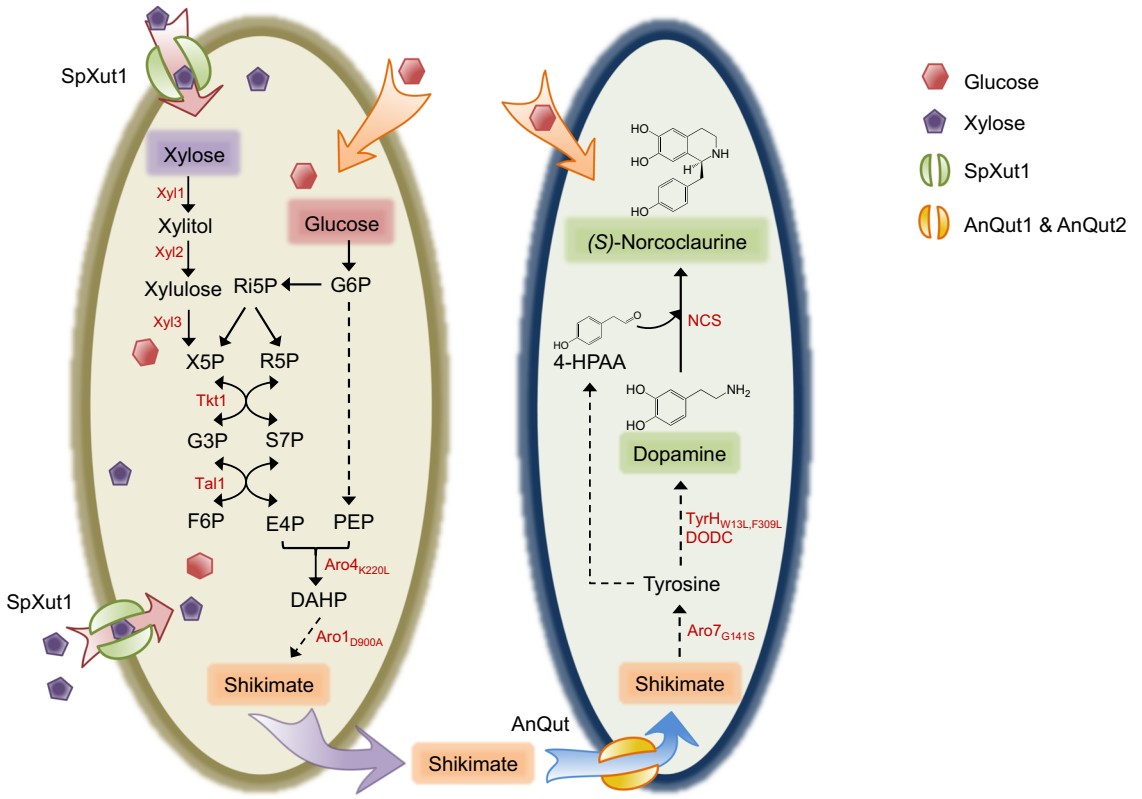

**Fig. 1 | Schematic of the *S. stipitis*-*S. cerevisiae* consortium that converts mixed sugars to (*S*)-norcoclaurine.** PEP phosphoenolpyruvate, G6P glucose-6-phosphate, Ri5P D-ribulose-5-phosphate, X5P D-xylulose-5-phosphate, R5P ribose-5-phosphate, G3P glyceraldehyde-3-phosphate, S7P sedoheptulose-7-phosphate, F6P fructose-6-phosphate, E4P erythrose-4-phosphate, DAHP 3-deoxy-D-arabinoheptulosonate 7-phosphate, 4-HPAA 4-hydroxyphenylacetaldehyde, Xyl1 xylose reductase, Xyl2 xylose dehydrogenase, Xyl3 xylulokinase, Tkt1

transketolase, Tal1 transaldolase, SpXut1 xylose transporter free from glucose inhibition from *Spathaspora passalidarum*, Aro4$_{K220L}$ feedback-resistant DAHP synthase mutant, Aro1$_{D900A}$ pentafunctional arom protein mutant, AnQut quinate permeases from *Aspergillus niger*, Aro7$_{G141S}$ feedback-resistant chorismate mutase mutant, TyrH$_{W13L,F369L}$ yeast active tyrosine hydroxylase mutant, DODC DOPA decarboxylase, NCS norcoclaurine synthase. Dashed arrows indicate that the conversion occurs via multiple steps.

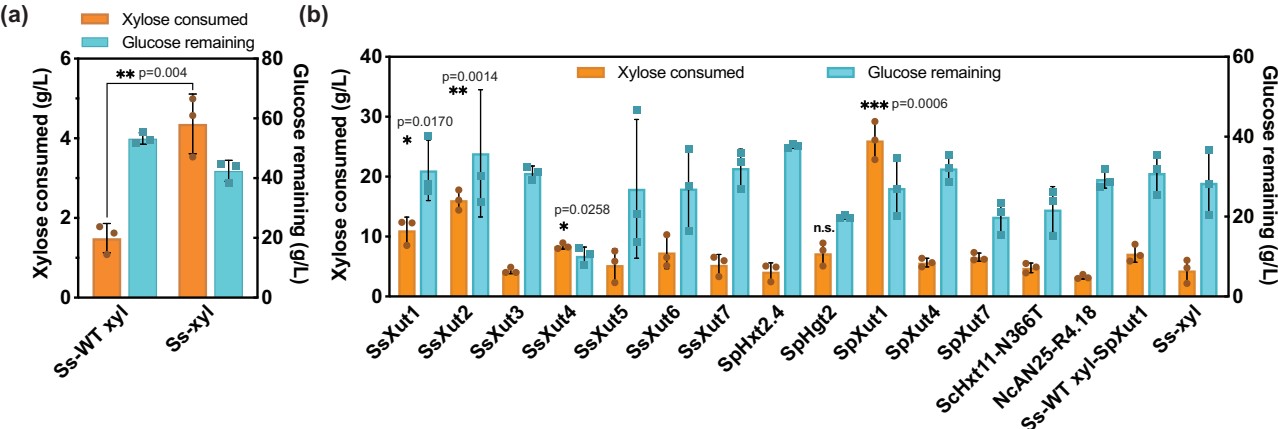

**Fig. 2 | Metabolic engineering of *S. stipitis* for efficient glucose/xylose co-utilization.** The SC-ura medium containing 70 g/l glucose and 40 g/l xylose was used. High concentrations of both sugars were used in HCDF considering that glucose at a regular concentration (e.g., 20 g/l) would be depleted in the first 24 h. It was therefore difficult to observe the impact of CCR on xylose assimilation. **a** Sugar utilization of strain Ss-WT xyl and Ss-xyl. **b** Screening of xylose-specific transporters cloned from different sources. Bars indicate xylose consumed and glucose remaining after the 5-day fermentation. Ss *S. stipitis*, WT wild type, Sp *Spathaspora passalidarum*, Sc *Saccharomyces cerevisiae*, Nc *Neurospora crassa*. Ss-xyl carries the engineered xylose assimilation pathway with the promoters of the five identified genes swapped with constitutive ones whereas Ss-WT xyl-SpXut1 contains the overexpressed SpXut1 transporter with the wild type xylose conversion pathway. The rest of the strains have both the overexpressed transporter and the engineered xylose assimilation pathway. Samples were collected after 120 h of cultivation. Data are presented as mean ± S.D., $n = 3$ per group. Statistical analysis for comparing xylose consumption was performed using a two-sided Student's *t* test. Selected comparisons are shown. *$p < 0.05$, **$p < 0.01$, ***$p < 0.001$ versus the strain Ss-xyl. Source data are provided as a Source Data file.

## Xylose transporters enhanced simultaneous mixed-sugar utilization

The genome of *S. stipitis* harbors seven annotated xylose transporters, encoded by *XUT1-7*[34]. We cloned each of these seven transporters downstream of a strong constitutive promoter into the pMG-xyl plasmid carrying the refactored xylose pathway. HCDF demonstrated that SsXut1, SsXut2, and SsXut4 improved xylose consumption to 8–16 g/l, with 34–60 g/l glucose being co-utilized. However, the other four transporters (i.e., SsXut3, and SsXut5-7) only enabled marginal co-sugar utilization (Fig. 2b). In our previous RNA-seq analysis, the transcription of *XUT1* was repressed by 16.7-fold at 15 h in the presence of glucose compared to the xylose-containing medium, while all other transporters including *XUT2* and *XUT4* were barely transcribed under all three culture conditions (Supplementary Table 1). The activation at the transcriptional level through promoter swapping confirms that SsXut1, SsXut2, and SsXut4 are all functional xylose transporters. The sugar utilization and cell growth profiles of the strain Ss-xyl expressing xylose transporters over a course of 120 h were summarized in Supplementary Fig. 3.

*S. stipitis* belongs to the CUG clade, which reassigns the CUG codon of leucine to serine. The vast majority of the CUG clade yeasts possess innate xylose-utilizing capability (Supplementary Table 2), long serving as reservoirs for isolating genes involved in xylose uptake and metabolism[35]. Among the CUG clade yeasts, *Spathaspora passalidarum* was reported as the lone yeast strain to date, natively capable of simultaneously co-utilizing glucose, xylose, and cellobiose[36]. Therefore, we hypothesized *S. passalidarum* possesses unique sugar transporters responsible for xylose uptake. Sequence alignment indicated that *S. passalidarum* contains three xylose transporters homologous to SsXut1, SsXut4, and SsXut7 from *S. stipitis*. In addition, previous transcriptome analysis of *S. passalidarum* demonstrated that the expression of two hexose transporters encoded by *HGT2* and *HXT2.4* were substantially higher in xylose-containing culture than in the glucose-containing culture[37]. We incorporated each of these five *S. passalidarum* transporters into the plasmid pMG-xyl and transformed the resulting plasmids individually into *S. stipitis* FPL-UC7. The best strain, Ss-xyl-SpXUT1, expressing Xut1 from *S. passalidarum*, co-utilized 26 g/l xylose and 43 g/l glucose in HCDF (Fig. 2b and Supplementary Fig. 3). We also overexpressed codon-optimized Hxt11-N366T from *S. cerevisiae*[38] and AN25-R4.18 from *Neurospora crassa*[39], both of which were previously reported to facilitate xylose transport into *S. cerevisiae* in mixed-sugar conditions (Fig. 2b); however, neither transporter improved xylose uptake in *S. stipitis*.

To assess whether all six genes in the plasmid pMG-xyl-SpXUT1 are necessary for efficient mixed-sugar utilization, we constructed another six plasmids by removing one gene at a time from pMG-xyl-SpXUT1 followed by individual plasmid transformation into *S. stipitis*. In contrast to the control strain harboring the complete set of six genes, removal of any gene from the combination resulted in a drastically reduced efficiency of mixed-sugar utilization, with xylose assimilation being particularly affected (Supplementary Fig. 4). Among them, the exclusion of the *XYL1* gene had a profound effect on overall growth, while the exclusion of the *SpXUT1* gene had the least impact. The removal of *XYL1* significantly hindered growth due to the reduced utilization of both glucose and xylose. A comparable pattern was observed when excluding *XYL2*, *XYL3*, or *TKT1*. This might be due to the impacts of an attenuated cofactor balance and/or the reoccurrence of CCR on central metabolism[40]. The exclusion of *SpXUT1* primarily resulted in a reduction in xylose utilization, leading to a comparatively less pronounced impact on overall growth compared to the other exclusions. Further investigation using techniques like [13]C-metabolic flux analysis is necessary to unravel the intricate relationship between sugar utilization and biomass production in these six variants. Nevertheless, the expression of the heterologous xylose transporter, SpXut1, in combination with promoter swapping of the genes involved in

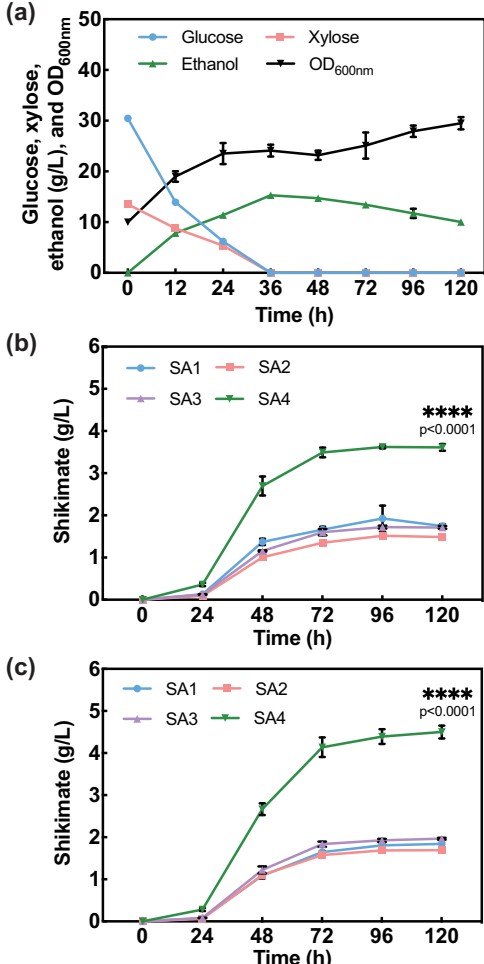

**Fig. 3 | The fermentation and shikimate production profiles after CCR relief. a** The fermentation profile of *S. stipitis* strain (Ss-xyl-SpXUT1) with CCR relief. Initial cell density ($OD_{600nm}$) was set at ~10. **b** Shikimate production by four engineered *S. stipitis* strains. For both (**a**) and (**b**), SC-leu-ura medium containing 28 g/l glucose and 12 g/l xylose was used. **c** Shikimate production by four engineered *S. stipitis* strains in SC-leu-ura medium containing 30 g/l glucose and 30 g/l xylose. Data are presented as mean ± S.D., *n* = 3 per group. Statistical analysis was performed using a two-sided Student's *t* test. ****$p < 0.0001$ versus SA1-3 strains. Source data are provided as a Source Data file.

xylose conversion alleviated CCR, enabled simultaneous assimilation of 28 g/l glucose and 12 g/l xylose within 36 h in *S. stipitis* (Fig. 3a). After depletion of the mixed sugars, the engineered strain started to consume ethanol for energy supply and biomass formation, resulting in a gradual increase in cell density from 48–120 h.

## CCR removal improved shikimate production in *S. stipitis*

To test the shikimate-producing capability after CCR mitigation, we constructed four shikimate-producing *S. stipitis* strains (SA1–4). SA1, SA2, and SA3 represent control strains, each carrying the previously constructed plasmid, pMG-SA (harboring the three-gene shikimate accumulation pathway for *S. stipitis*[23]), along with one of the following three plasmids, pMG-xyl, pMG-SpXUT1, and pMG-eGFP (equivalent to an empty vector), respectively (Supplementary Fig. 2). All three control strains accumulated extracellular shikimate at low levels, ranging from 1.52 to 1.92 g/l in SC-leu-ura medium containing 28 g/l glucose and 12 g/l xylose. In contrast, the SA4 strain harbors both pMG-xyl-SpXUT1 and pMG-SA, produced 3.62 g/l shikimate under the same culture condition, 89–138% higher than the control strains (Fig. 3b). To validate the hypothesis that increasing xylose concentration in the medium could

further increase flux toward E4P and subsequently improve shikimate production, the four strains (SA1–4) were cultured in SC-leu-ura containing 30 g/l glucose and 30 g/l xylose. This yielded the maximal shikimate titer (4.50 g/l, Fig. 3c), which is 80% higher than the best *S. cerevisiae* shikimate-producing strain that has undergone markedly more extensive engineering[41].

### Discovery of unique transporters indispensable to shikimate uptake in *S. cerevisiae*

Efficient translocation of the connecting molecule(s) is a prerequisite for building an efficient consortium. Shikimate, as an endogenously produced metabolite, can be efficiently secreted into the medium. Therefore, as an early attempt, we tested *S. cerevisiae* for its shikimate import capability. To enhance shikimate conversion upon the uptake, we constructed a few recombinant strains overexpressing *E. coli* shikimate kinase AroL, the endogenous penta-functional enzyme Aro1, and chorismate mutase variant Aro7$_{G141S}$, individually or in combination. These enzymes were previously reported to benefit shikimate conversion and consequently enhance the metabolic flux entering the downstream conversion pathways, such as tyrosine biosynthesis (Fig. 4a)[42–44]. Unfortunately, none of the resulting strains were capable of importing and consuming shikimate (Fig. 4b). We subsequently cloned two previously characterized prokaryotic shikimate transporters, *E. coli* ShiA[45] and *Corynebacterium glutamicum* ShiA[46], with their

codons optimized, but found that neither of them improved shikimate uptake in *S. cerevisiae* (strains EcShiAco and CgShiAco in Fig. 4c). In a previous study of cytosolic NADPH/NADP$^+$ ratio in *S. cerevisiae*, shikimate transport was suggested as a symport, with a very slow shikimate uptake rate of 0.005 μmol per second per gram of dry cell weight[47]. Therefore, it was imperative to identify additional eukaryotic shikimate importers that can be functionally expressed in *S. cerevisiae*.

A literature search as far back to the 1970s revealed that three fungi, *Aspergillus niger*, *Aspergillus nidulans*, and *Neurospora crassa*, possess shikimate-assimilating capabilities[48,49]; however, no specific studies regarding shikimate uptake in eukaryotes were published yet. We grew these three fungi in minimal media containing shikimate as the sole carbon source for 5 days. *A. niger* grew markedly better than *A. nidulans* and *N. crassa* (the latter two barely grew), strongly indicating the existence of a highly functional shikimate import system in *A. niger*. Despite the poor growth in the current culture condition, *A. nidulans*[50] and *N. crassa*[51] were reported to transport quinate, whose structure is similar to shikimate (Supplementary Fig. 5), replacing the alkenyl group in shikimate with a hydroxyalkyl group. Furthermore, there are also quinate/shikimate dehydrogenases that accept shikimate and quinate as substrates using the same binding site[52,53]. Based on this thread of evidence, we hypothesized that some quinate permeases in fungi might be capable of transporting shikimate.

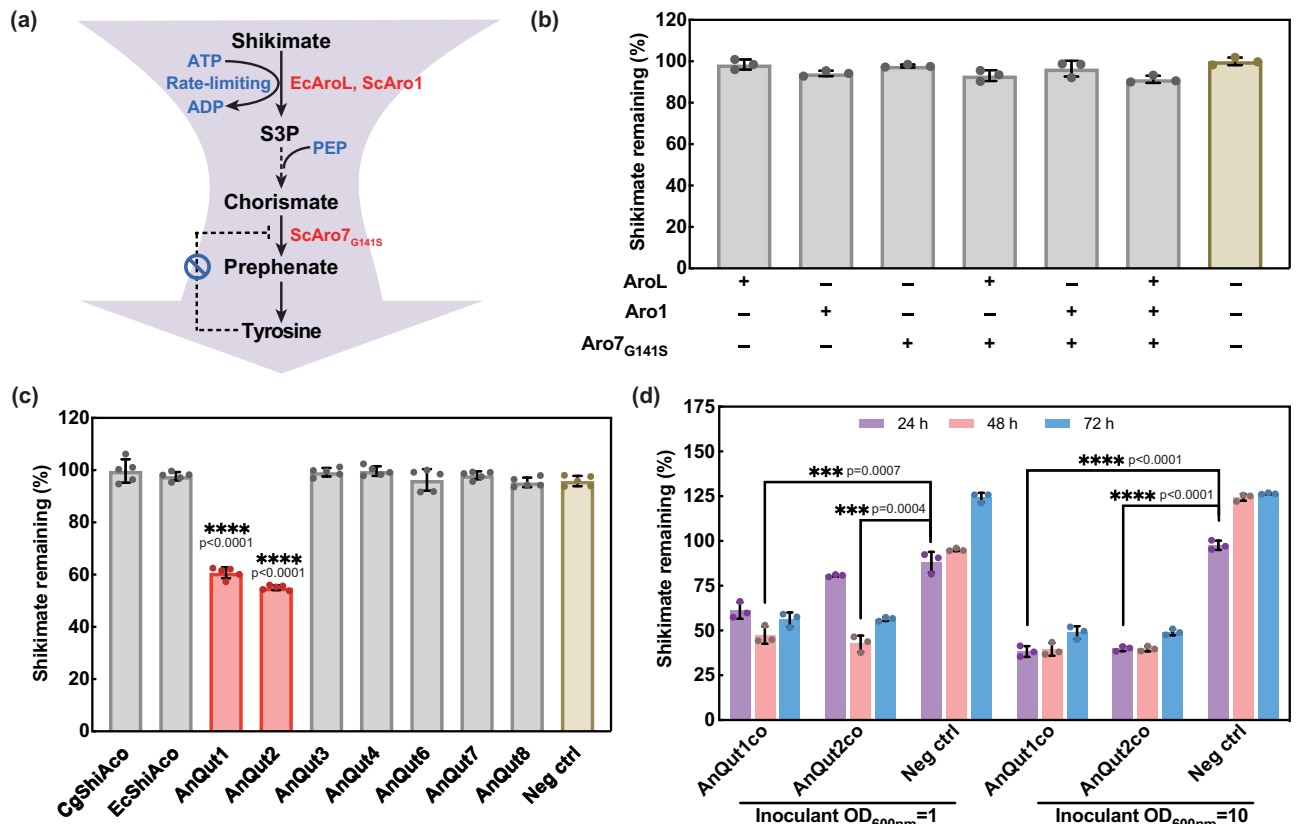

**Fig. 4 | Screening of transporters for efficient shikimate uptake by *S. cerevisiae*. a** The shikimate-to-tyrosine pathway. **b** Shikimate utilization by *S. cerevisiae* strains with overexpression of Aro7$_{G141S}$, AroL, and/or Aro1 in the medium containing 0.5 g/l shikimate after 3-day fermentation. **c** Shikimate utilization of *S. cerevisiae* strains containing prokaryotic shikimate transporters and eukaryotic quinate permeases in the medium containing 1 g/l shikimate after 3-day fermentation. **d** Shikimate utilization by *S. cerevisiae* strains harboring the codon-optimized *A. niger* quinate permeases, AnQut1co and AnQut2co. The negative control strains contained the empty plasmid, pRS413. Strains were grown in SC-his medium containing 80 g/l glucose and 1 g/l shikimate. The negative control samples

demonstrated that higher than 100% of shikimate remained in the medium at the end of the fermentation, which was the consequence of secretion of the endogenously produced shikimate in combination with reduced volumes due to minor evaporation. Cg *Corynebacterium glutaminum*, Ec *Escherichia coli*, An *Aspergillus niger*, co codon optimization. Data are presented as mean ± S.D., $n = 3$ per group for (**b**) and (**d**) and $n = 5$ per group for (**c**). Statistical analysis was performed using a two-sided Student's $t$ test. Selected comparisons are shown. ***$p < 0.001$, ****$p < 0.0001$ versus negative controls. Source data are provided as a Source Data file.

The first reported quinate permease among these three fungi is Qa-Y from *N. crassa*[51]. We ran a BLAST of Qa-Y against the non-redundant protein sequence database of *A. niger*. A sequence (XP_001392502.1, designated as AnQut1) with the maximal similarity (76%) to Qa-Y was identified and used as the second query to re-blast the *A. niger* protein database. Seven additional putative quinate transporters (designated as AnQut2-8) were found to share similarities ranging from 50 to 78% (Supplementary Table 3). As all eight DNA sequences encoding putative quinate permeases harbor introns, we cloned the genes using cDNA libraries synthesized from *A. niger* cultivated in medium with shikimate as the sole carbon source. Reverse transcription-PCR revealed the coding regions of seven putative quinate transporters (AnQut7 was not successfully amplified), which were subsequently cloned into *S. cerevisiae* for the shikimate uptake assay. Unlike *A. niger*, *S. cerevisiae* cannot efficiently use shikimate as the sole carbon source natively. Therefore, the uptake assay was performed in the presence of sugar, mimicking the culture conditions for the desired consortium. Shikimate was added to the medium at a concentration of 1 g/l instead of 4.5 g/l, the maximal titer achieved by the optimized *S. stipitis* monoculture. This was because we postulated that shikimate produced by *S. stipitis* would not instantaneously reach a high level, and in the consortium, *S. stipitis* would likely be added to represent a smaller population than *S. cerevisiae* due to the more challenging expression issues with the downstream heterologous enzymes. As a result, unlike the other five putative quinate permeases, AnQut1 and AnQut2 displayed substantially higher shikimate import activity, with the corresponding strains consuming as much as 45% shikimate after a 72-h fermentation (Fig. 4c).

To further improve shikimate transport, the previous manipulations (e.g., overexpression of EcAroL, ScAro1, and/or ScAro7$_{G141S}$) were performed in the *S. cerevisiae* strain overexpressing AnQut1co to facilitate the conversion of shikimate to downstream products. Unexpectedly, their incorporations imposed marginal or even impaired shikimate conversion (Supplementary Fig. 6a). When EcAroL or ScAro1 was overexpressed, the shikimate conversion was strongly interrupted. Further cell growth tests revealed that overexpression of AroL or Aro1 in *S. cerevisiae* strains containing AnQut1co impeded cell growth in the presence of shikimate (Supplementary Fig. 6b). We postulated that overexpression of AnQut1co and shikimate kinase (encoded by AroL or Aro1) enabled shikimate import and increased the conversion of shikimate to shikimate-3-phosphate (S3P) in a reaction that uses ATP as a cofactor (Fig. 4a). It is possible that the excessive drainage of the energy molecule in the presence of high-level shikimate led to poor cell propagation. Alternatively, whether the accumulation of certain downstream intracellular metabolite(s) resulting from overexpressed shikimate kinase may have inhibited cell growth remains as another possibility to be investigated in the future. Hence, only Aro7$_{G141S}$ along with the codon-optimized version of AnQut1 or AnQut2 was expressed in *S. cerevisiae*, which resulted in shikimate conversions ranging from 53 to 57% at 48 h during low cell-density fermentation (LCDF) and 60 to 62% at 24 h during HCDF (Fig. 4d). The indistinction between LCDF and HCDF in terms of shikimate import could be attributed to the fact that the cell density of *S. cerevisiae* only takes ~140 min to double in the synthetic medium, which means that it takes only about three generations to reach an OD$_{600nm}$ of 10 from an OD$_{600nm}$ of 1. As a result, no significant difference was observed between LCDF and HCDF for shikimate import after sampling every 24 h. To the best of our knowledge, these quinate permeases from *A. niger* represent two unique shikimate transporters functionally expressed in *S. cerevisiae*.

## (S)-norcoclaurine production by the yeast co-culture

BIAs are a structurally diverse family of plant-specialized L-tyrosine-derived compounds with more than 2500 known structures. As the first committed backbone intermediate in the BIA pathway, (*S*)-

norcoclaurine is synthesized via the condensation of dopamine and 4-hydroxyphenylacetaldehyde (4-HPAA) catalyzed by norcoclaurine synthase (NCS) (Fig. 1). Both dopamine and 4-HPAA are derived from a common precursor, L-tyrosine. In yeasts, 4-HPAA is produced endogenously via the Ehrlich pathway, whereas dopamine is exogenous and formed through sequential oxidation and decarboxylation catalyzed by tyrosine hydroxylase (TyrH) and L-3,4-dihydroxyphenylalanine decarboxylase (DODC), respectively. Therefore, in this study the native enzymes involved in the tyrosine-to-HPAA conversion remained untapped, and efforts were devoted to increasing the flux to dopamine. A copy of TyrH*-DODC-NCS was cloned into pRS413-AnQut1co-Aro7*, pRS415, and pRS416 plasmids in parallel, and co-transformed into the strain CEN.PK2-1c[54], resulting in a strain NC1 with one copy of AnQut1co and Aro7*, and three copies of TyrH*, DODC, and NCS. Here TyrH* carries two mutations (W13L and F309L) that substantially increase the tyrosine hydroxylase activity while reducing the undesired DOPA oxidase activity[15]. In the pretest, the NC1 strain was cultivated in 2 × SC-his-leu-ura medium supplemented with 1 g/l SA, yielding 2.8 mg/l (*S*)-norcoclaurine and consuming 0.66 g/l SA with a starting cell density of 0.2 (Supplementary Fig. 7). To improve (*S*)-norcoclaurine production and increase shikimate conversion, higher initial cell densities of 1, 5, and 10 were applied to monoculture fermentation. Samples were collected after 96 h of fermentation. Results revealed that a complete conversion of shikimate and the titer of (*S*)-norcoclaurine at 3.1 mg/l were obtained when the starting cell density was five.

In a preliminary experiment, co-cultures of *S. stipitis* SA4 and *S. cerevisiae* NC1 conducted with various Ss:Sc inoculating ratios (i.e., 9:1, 5:1, 3:1, 1:1, 1:3, 1:5, and 1:9) and an initial total cell density of 0.2–0.3 in 2 × SC medium containing 1 g/l SA and 5 g/l L-ascorbic acid (stabilizing the tetrahydrobiopterin cofactor of tyrosine hydroxylase[18]) yielded (*S*)-norcoclaurine with titers ranging from 0.17 to 1.7 mg/l (Supplementary Fig. 8). Increasing the representation of NC1 benefited the production when the Ss:Sc ratio was high (from 9:1 to 3:1). The production from the ratio of 3:1 (Ss:Sc) was not significantly different from the one obtained with the ratio of 1:3 (Ss:Sc), whereas further increasing the representation of NC1 adversely affected (*S*)-norcoclaurine production (from 1:3 to 1:9). This may be due to the low cell density of the SA4 strain, which resulted in insufficient shikimate supply for the downstream conversion occurring in the NC1 strain. Considering the potential benefit of a higher cell density of NC1 as suggested in the monoculture fermentation (Supplementary Fig. 7), a different inoculation strategy was tested without fixing the initial total cell density: The initial cell density of SA4 was kept at 0.2–0.3 whereas NC1 was inoculated with an initial density ranging from 0.2 to 12 (Fig. 5a). This strategy resulted in increases in (*S*)-norcoclaurine production, with the titer ranging from 3.2 to 12.0 mg/l. The maximal titer was achieved when NC1 was inoculated with an initial OD$_{600nm}$ of 3, representing a six-fold increase over the maximal titer achieved with an initial total cell density control (1.7 mg/l in Supplementary Fig. 8). We also quantified cell density, accumulation of dopamine and shikimate, and residual glucose and xylose in the cocultures (Supplementary Fig. 9). Consistent with the optimized titer yielded when NC1 was inoculated with an initial OD$_{600nm}$ of 3, an ignorable level of shikimate was detected. Lower amounts of inoculated NC1 resulted in high levels of shikimate accumulation whereas higher initial amounts of NC1 resulted in undesired buildups of xylose.

In addition to the inoculated amount of the strain NC1, another variable is the growth stage of the strain SA4. Considering the potential benefit of allowing SA4 to grow first and accumulate shikimate to a decent level prior to introducing NC1, we tested a sequential inoculation strategy. Specifically, NC1 was inoculated at a cell density of three into the SA4 culture that had been cultivated for a varying period ranging from 0 to 48 h. The result showed that delaying the addition of NC1 resulted in drastic decreases in the (*S*)-norcoclaurine titer from

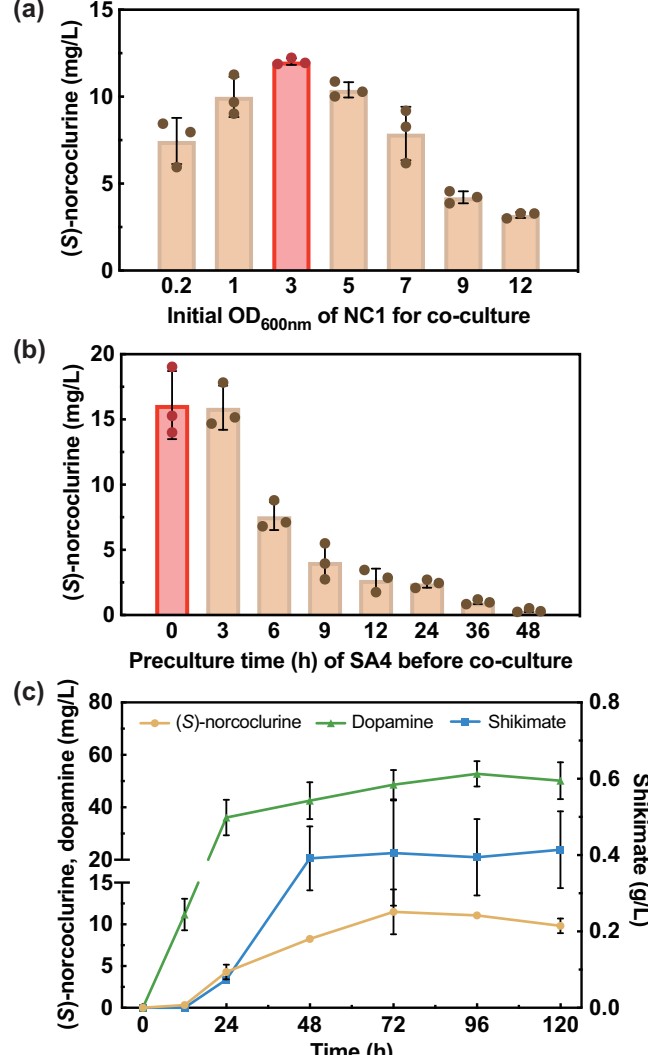

**Fig. 5 | Optimizing (S)-norcoclurine production by yeast consortia. a** (S)-nor-coclurine titers of the consortia inoculated with different initial *S. cerevisiae* NC1 cell densities (OD$_{600nm}$). The initial cell density of *S. stipitis* SA4 was kept at 0.2–0.3 and the two strains were simultaneously introduced into the culture. Samples were collected after 96 h of fermentation. **b** (S)-norcoclurine titers of the consortia with a sequential inoculation strategy. *S. stipitis* SA4 was first grown in fermentation medium (2x SC medium containing 30 g/l glucose, 30 g/l xylose, and 5 g/l L-ascorbic acid). After 0, 3, 6, 9, 12, 24, 36 and 48 h, *S. cerevisiae* NC1 was introduced with an OD$_{600nm}$ of three into the corresponding *S. stipitis* SA4 culture. Samples were collected after 96 h of fermentation. **c** The production time course of (S)-norcoclurine, dopamine, and shikimate in the consortium under the optimal fermentation condition. The initial cell density of SA4 was 0.2–0.3 and NC1 was simultaneously introduced into the culture with an OD$_{600nm}$ of three. Samples were collected every 24 h post the starting of the co-culture. Data are presented as mean ± S.D., *n* = 3 per group for (**a**) and (**b**) and *n* = 5 per group for (**c**). Source data are provided as a Source Data file.

16.1 mg/l (at 0 h) to 0.32 mg/l (at 48 h) (Fig. 5b), suggesting that early inoculation of NC1 was more effective in achieving (S)-norcoclurine production. The delayed inoculation of NC1 led to an increased xylose consumption and an undesired shikimate buildup by SA4, particularly if SA4 was pre-cultured for 36 and 48 h. Consequently, there were insufficient nutrients in the medium available for NC1, which significantly reduced (S)-norcoclurine titer (Fig. 5b and Supplementary Fig. 10).

The production time course of (S)-norcoclurine and dopamine and the concentrations of the residual sugars, shikimate, and other

byproducts were determined (Fig. 5c and Supplementary Figs. 11 and 12). Under the optimized condition of simultaneously mixing NC1 and SA4 with initial OD$_{600nm}$ of three and 0.2–0.3, respectively, the maximal (S)-norcoclurine titer of the Ss/Sc consortium (11.5 mg/l) was obtained. This was almost 110-fold and 37-fold higher than those reported in the literature for a *S. cerevisiae* monoculture[15] and a *S. stipitis* monoculture[55], respectively, supporting a strong synergy between the two species in the consortium. In addition, more than three-fold higher accumulation of dopamine was observed during the coculture. This is consistent with the previous monoculture studies in *S. cerevisiae*[15] and *S. stipitis*[55] that reported dopamine accumulation, suggesting that either low *NCS* expression level or insufficient 4-HPAA precursor supply limited (S)-norcoclurine formation. The native Ehrlich pathway could also play a role as it might rapidly transform 4-HPAA to the corresponding fusel acid, 4-hydroxyphenylacetic acid (4-HPAC) or alcohol tyrosol[1]. It was also observed that the mixed sugars (including ~20 g/l glucose and ~20 g/l xylose) were rapidly converted to the main side product ethanol and a minor amount of glycerol via the Crabtree-positive yeast NC1, reaching the maximal levels at 12 h (Supplementary Fig. 12). From 12 to 48 h, ethanol and glycerol were consumed along with xylose and residual glucose to continuously support energy metabolism, biomass formation, and production of the target compound, its intermediates, and acetate as a byproduct.

## Discussion

A key discovery in this study is the identification of unique transporters. For xylose transporter, Young et al. previously identified a conserved motif (G-G/F-XXX-G) enriched among multiple heterologously expressed monosaccharide transporters that conferred faster growth rates on xylose than on glucose[56]. Farwick et al. studied the native hexose transporter GAL2 in *S. cerevisiae* and found that a single point mutation N376F completely abolished glucose transport activity while still enabled effective xylose translocation[57]. To understand if these critical residues exist in the transporters in the current study, the amino acid sequences of the best transporter, SpXUT1, and all seven SsXUT transporters were aligned to GXS1 from *Candida intermedia* (named CiGXS1)[56] and ScGAL2[57] (Supplementary Data 2 and 3). Interestingly, four out of the eight transporters were found to harbor the G-G/F-XXX-G motif, supporting that the conserved motif is highly enriched in the transporters that are functional for xylose. Moreover, among the eight, SpXUT1, SsXUT1, SsXUT2, and SsXUT4 contain methionine, methionine, phenylalanine, and tyrosine at the counterparts of the ScGAL2$_{N376}$ position. Based on Farwick's study, ScGAL2$_{N376Y}$ was demonstrated to be free of glucose inhibition although the xylose transport activity was also negatively impacted whereas ScGAL2$_{N376F}$ enabled a strong and glucose-resistant transport of xylose. It is worth pointing out that both Young and Farwick's studies were based on *S. cerevisiae* EBY.VW4000, a widely used platform strain with at least 20 endogenous hexose transporters deleted for characterization of additional sugar transporters[58]. Such a parental strain is not easily available for *S. stipitis* because sequentially knocking out more than 20 native transporters in combination of selection marker recycling is still challenging due to the dominancy of the inherent nonhomologous end joining mechanism that prevents precise genome editing. Therefore, in the current yeast consortium study although SpXUT1, SsXUT1, SsXUT2, and SsXUT4 can effectively transport xylose in the presence of glucose, we cannot exclude their activities toward glucose. Additional validation such as sugar uptake kinetics experiments in a similar strain background to *S. cerevisiae* EBY.VW4000 will be needed for detailed transporter characterization studies in the future.

One determinant of labor division in synthetic consortia involves effective metabolite transfer between sub-populations. It is crucial to carefully identify an appropriate connecting molecule to distribute

pathways when designing the consortia at the initial stage. Although some pathway intermediates (e.g., taxanene[25], butyrate[59], tyrosine, and *p*-coumaric acid[28]) can be ingested by microorganisms natively, the re-assimilation of other connecting molecules is impeded by the lack of efficient transportation systems. For example, because of the low activity of the heterologous *cis*, *cis*-muconic acid (MA) synthesis pathway, 3-dehydroshikimic acid (DHS) was accumulated extra-cellularly in several monocultures that produce MA[60,61]. A stable *E. coli* co-culture system was constructed to facilitate the production of MA[26], in which one strain contained the upstream module that converted xylose to the intermediate, DHS, and the other strain, which was glu-cose-dependent, harbored the downstream module that converted DHS to MA. The discovery of a DHS transporter enabled the re-assimilation of extracellular DHS, which resulted in increased yield (0.35 g $_{MA}$/g $_{sugar}$). In our current study, the entire BIA precursor synthesis pathway was split at the shikimate node. At the initial stage of the project, tyrosine was also being considered as a potential con-necting molecule. We did not select it further due to the initial concern of the potential competition with protein synthesis and later found that the low solubility of tyrosine in water (0.5 g/l) at room tempera-ture and neutral pH[62] presents a practical challenge in the pre-test experiments related to supplementation of a high-concentration pre-cursor. Alternative methods, such as separately feeding a high-concentration tyrosine stock solution prepared in 1 N HCl into the fermentation medium, have been attempted. Unfortunately, this approach has resulted in the inhibition of cell growth due to low pH, leading to low (*S*)-norclaurine productions (Supplementary Fig. 13). Therefore, we switched to the highly soluble endogenous metabolite, shikimate, overcame the hurdle of its import by identifying two unique eukaryotic shikimate transporters, and proved their indispensable roles in achieving efficient translocation of shikimate from the medium into the intracellular space of *S. cerevisiae*.

The fabrication of synthetic microbial consortia is an emerging design paradigm to produce high-value chemicals. The advantages of using a consortium are two-fold: (1) Partitioning a complex metabolic pathway in different specialists is beneficial for achieving a plug-and-play design with optimal function, and (2) Dividing long pathways into two members of the consortium can not only reduce metabolic stress but also offer a perspective to balance the ratio of the two modules. For (1), Zhou et al. built a mutualistic consortium for the efficient production of oxygenated terpenoids, which, also known as iso-prenoids, are a large and diverse class of natural products with various applications. To overcome the inability of *E. coli* to functionally express cytochrome P450s (CYPs), a rapid taxadiene-producing *E. coli* was combined with *S. cerevisiae* that is amenable to the expression of CYPs. After a series of optimization, the microbial consortium produced 33 mg/l oxygenated taxanes, which could barely be detected in either *E. coli* or *S. cerevisiae* monoculture that contained the entire pathway[25]. Following a similar rationale, an *E. coli* consortium was developed and optimized to efficiently produce flavan-3-ols, leading to a 970-fold improvement in titer over the monoculture[27]. For (2), previous success includes a naringenin-producing *E. coli*/*E. coli* consortium that accu-mulated six-fold more naringenin than the monoculture control[28]. The improved production was attributed to three factors: alleviated metabolic burden, host selection based on their metabolic character-istics, and flexible pathway balancing. In our study, *S. stipitis* was employed because of its superior xylose-assimilating capability and active pentose phosphate pathway, while *S. cerevisiae* was recruited because of the previous success in addressing the hardships to express challenging enzymes and produce plant-sourced secondary metabo-lites. Overall, the selection of well-suited expression environments according to the functionality of pathway enzymes led to the final high product titer.

As the majority of the synthetic microbial consortia are complex with dynamic compositions, one of the key factors in their design is to prevent the dominance of one species over another due to a shorter doubling time or an advantage of utilizing certain substrates. To achieve a stable coexistence among the species, conventional strate-gies such as titration of the inoculum ratio and optimization of mixing time can be exploited. In our study, the Ss/Sc consortium yielded the maximal (*S*)-norcoclaurine titer (11.5 mg/l) when these two determi-nants were optimized simultaneously (i.e., an initial OD$_{600nm}$ of three for NC1 and 0.2–0.3 for SA4; simultaneous inoculation of the two seed cultures at 0 h). Additionally, considering the best inoculation strategy so far still led to an accumulation of shikimate at 0.41 g/l after 120 h of fermentation (Fig. 5), we sought to further increase (*S*)-norcoclaurine production by increasing the copy number of shikimate transporter gene AnQut1co. Two more strains consisting of two-copy and three-copy AnQut1co were constructed by incorporating AnQut1co into pRS415-TyrH*-DODC-NCS (named NC2) and then into pRS416-TyrH*-DODC-NCS (named NC3) (Supplementary Data 1). A strain named Sc-pRS356 containing empty plasmids pRS413, pRS415, and pRS416 was included as a negative control. Unexpectedly, doubling and tripling the dose of AnQut1co significantly decreased the (*S*)-norcoclaurine titer from 5.7 mg/l (NC1) to 1.9 mg/l (NC2) and 1.3 mg/l (NC3) (Sup-plementary Fig. 14). Correspondingly, decreases in cell density and higher contents of residual shikimate were observed in the cocultures of SA4/NC2 and SA4/NC3 (Supplementary Fig. 15). This result might be correlated with the previous observation of significantly reduced growth rates of the strains overexpressing shikimate kinase (i.e., EcAroL or ScAro1) (Fig. 4a and Supplementary Fig. 6). To study the causality, future efforts should be first focused on investigating the ATP drainage in these variants.

In parallel, we also attempted to measure the population compositions of the SA4/NC1 consortium using their phenotypic difference that only *S. stipitis* enables the catabolism of xylose. We collected consortium samples every 24 h, diluted 10–1000 folds, and plated them on three different agar plates that allowed only SA4 (SC-Leu-Ura + 2% xylose) or both SA4 and NC1 (SC-His-Leu-Ura + 2% glucose and YPA + 2% glucose) to survive. However, we were unable to accurately ascertain the ratio of the consortium members over time presumably due to plasmid instability. The numbers of colonies appearing on the YPA + 2% glucose plates were 2.6–13.6-fold higher than those on the SC-His-Leu-Ura + 2% glucose plates. To address this challenge, genomic integration of the pathway needs be performed in the future to achieve higher genetic stability. Additionally, accurately monitoring population dynamics in real time can be achieved using multi-color flow cytometry analysis[63,64] that involves quantifying different populations with corresponding integrated fluorescent proteins-encoding genes.

Lastly, for industries to eventually utilize microbial consortia for chemical production, self-regulation is a key feature desired to be established in co-cultures because manual titration of cell densities of individual species will be challenging along with the scale-up pro-cesses. To this end, relieving the competition by using distinct carbon sources and establishing symbiotic relationships have been applied to improve the consortium stability and robustness[65-68]. Recently, a multi-metabolite crossing feeding strategy was developed to strengthen the connection between species. Relying on amino acid anabolism, energy metabolism, and a caffeate-responsive biosensor, autonomous reg-ulation of strain ratios was achieved in a three-strain co-culture for de novo biosynthesis of silybin and isosilybin[69]. Such a strategy is highly valuable for designing synthetic consortia that involve distinct species with different natural habitats including various substrate spectrums, metabolic performances, and metabolite profiles[70]. In all cases, reliable modeling tools to predict the interactions in microbial consortia that span different spatiotemporal scales combined with high-resolution multi-omics analysis are highly desired to harness the full potential of microbial consortia for large-scale industrial production[71].

In summary, a synthetic microbial consortium was constructed for the efficient production of a BIA pathway precursor. The long metabolic pathway was split into two modules and allocated to *S. stipitis* and *S. cerevisiae* to leverage the unique advantage of each host. One xylose transporter enhanced simultaneous glucose/xylose co-utilization, resulting in high-level shikimate production in *S. stipitis*. Two quinate permeases enabled the efficient translocation of SA to the *S. cerevisiae* cytoplasm from the medium. Subsequent optimization of the co-culture system yielded (*S*)-norcoclaurine at 11.5 mg/l, which is almost 110-fold and 37-fold higher than that of the *S. cerevisiae* monoculture and the *S. stipitis* monoculture, respectively. Through this case study, we have demonstrated the potential of consortia in producing compounds from complicated long pathways and identified transport of the connecting molecule as a critical design component to expedite the establishment of an effective consortium.

## Methods

### Strains, growth media, and materials

*S. cerevisiae* YSG50 was used for in vivo plasmid construction via DNA assembler[72]. *E. coli* WM1788 or DH5α was used to enrich all the plasmids. *S. stipitis* FPL-UC7 was a gift from the emeritus professor Thomas W. Jeffries (University of Wisconsin, Madison, WI)[33]. *S. stipitis* FPL-UC7 *leu2Δ* was derived from *S. stipitis* FPL-UC7 via the deletion of the gene encoding Leu2. *S. cerevisiae* BY4741 strain (*MAT*a *his3Δ1 leu2Δ0 met15Δ0 ura3Δ0*) was used for shikimate transporter screening and characterization. *S. cerevisiae* CEN.PK2-1C (*MAT*a *ura3-52 trp1-289 leu2-3,112 his3Δ1 MAL2-8C SUC2*) was used for (*S*)-norcoclaurine production. All wild-type yeast strains were grown in YPAD media containing 10 g/l yeast extract, 20 g/l peptone, 100 mg/l adenine hemisulfate, and 20 g/l glucose unless otherwise noted. Yeast transformants were grown in synthetic complete (SC) media supplemented with amino acid mix (MP medicals, San Diego, CA) according to the manufacturing guidance, adenine hemisulfate, and appropriate carbon sources. All yeasts and fungi were cultured at 30 °C with orbital shaking at 250 rpm unless otherwise noted. *E. coli* strains were selected in Luria-Bertani (LB) liquid media containing 100 mg/l ampicillin at 37 °C. All strains used in this study were stored in 25% glycerol at −80 °C.

### RNA isolation and cDNA library synthesis

*A. niger*, *A. nidulans*, and *N. crassa* were inoculated to 250 ml baffled shake flasks containing 20 ml minimal media plus 1 g/l shikimate as the sole carbon source for 5-day growth. Since *A. niger* grew markedly better than the other two, it was chosen as the repository for isolating shikimate transporter genes. The mycelia of *A. niger* were collected and frozen at −80 °C for total RNA extraction. Briefly, the mycelia of *A. niger* were ground into powder in liquid nitrogen, followed by the isolation of total RNA using RNeasy Mini Kit (Qiagen, Valencia, CA) and the removal of genomic DNA contamination using Turbo DNA-free Kit (Life Technologies, Carlsbad, CA). The purified total RNA was then reversely transcribed into a cDNA library using RevertAid First Strand cDNA Synthesis Kit (Thermo Scientific, Waltham, MA).

### Plasmid construction and yeast transformation

All the strains, vectors, and the primers used in this study are listed in Supplementary Data 1 and 4. All primers were synthesized by either Integrated DNA Technology (Coralville, IA) or DNA facility (ISU, Ames, Iowa). All restriction enzymes were purchased from Thermo Fisher Scientific (Waltham, WA) or New England Biolabs (Ipswich, MA). All plasmids were constructed by transforming strain YSG50 via the DNA assembler method[73,74] unless otherwise noted. The transformation into *S. cerevisiae* was carried out via LiAc/Ss carrier DNA/PEG method[75], while *S. stipitis* transformation was achieved by electroporation. After transformation, cells were grown on corresponding selective plates

until colonies appeared. The genomic DNA (gDNA) molecules of freshly grown *S. stipitis* FPL-UC7, *S. passalidarum*, and *S. cerevisiae* BY4741 were extracted using Wizard Genomic DNA Purification Kit (Promega, Madison, WI). The gDNA of *Candia tropicalis*, *Candida parapsilosis*, *Candida guilliermondii*, *Candida tenuis*, and *Debaryomyces hansenii* was a gift from Dr. Huimin Zhao at the University of Illinois Urbana Champaign. DNA fragments corresponding to *XYL1*, *XYL2*, *XYL3*, *TAL1*, *TKT1*, shortened promoters, and terminators were individually amplified using the gDNA of *S. stipitis* FPL-UC7 as the template. All the cassettes (SsADH1p-SsTEF1t, SsENO1p-*XYL1*-SsGLN1t, SsGLN1p-*XYL2*-SsAOX1t, SsOLE1p-*XYL3*-SsUAGt, SsPIR1p-*TKT1*-SsOLE1t, SsTEF1p-*TAL1*-SsADH1t) were constructed via overlap extension PCR (OE-PCR)[76], and assembled into ARS/CEN5-750bp plasmid backbone[77] to yield the pMG-xyl plasmid. Two restriction sites, AscI and XmaI, were added between the ADH1p and TEF1t to facilitate the rapid cloning of transporters. Codon-optimized *HXT11-N366T*[38] and *AN25-R4.18*[39] were prepared through GeneArt Gene Synthesis (Thermo Fisher Scientific, Carlsbad, CA). The DNA fragment encoding each of the candidate xylose transporters was amplified and co-transformed with the pMG-xyl plasmid linearized by AscI and XmaI (New England Biolabs, Ipswich, MA) for vector assembly. All yeast plasmids were isolated using Zymoprep Yeast Plasmid Miniprep II (Zymo Research, Irvine, CA) followed by transformation to *E. coli* for plasmid propagation. QIAprep Spin Miniprep Kit (Qiagen, Valencia, CA) was used for *E. coli* plasmid isolation. After verified by restriction enzyme digestion and Sanger sequencing, correct constructs were electroporated into *S. stipitis*. The transformants were selected on selective plates until colonies appeared. Vector pMG-SA was derived from the previously reported pMG-SA7.3 through the replacement of *URA3* selection marker with the *LEU2* marker.

All genes encoding quinate transporters were PCR amplified using the cDNA library of *A. niger* as the template, and ScADH1p and ScCYC1t were cloned using the genomic DNA of BY4741 as the template. These three fragments were purified using Zymo Gel DNA Recovery Kit (Zymo Research, Irvine, CA) followed by the assembly to pRS413 linearized by ApaI and SalI using Gibson Assembly Cloning Kit (New England Biolabs, Ipswich, MA). After preliminary screening, the best shikimate transporters, designated as AnQut1 and AnQut2, were codon-optimized and cloned into the multiple cloning sites of pRS413 and pRS416 along with ScADH1p and ScCYC1t. To increase the metabolic flux to the aromatic amino acid synthesis pathway, plasmids containing expression cassettes (ScADH1p-*AnQut1co*/*AnQut2co*-ScCYC1t, ScPGK1p-*ScAro7*$_{G141S}$-TDH2t, ScTEF1p-*EcAroL*-ScHXT7t, and ScPGK1p-*ScAro1*-ScHXT7t) were built. The plasmid, p2249 (a gift from Dr. John Deuber, University of Berkeley, CA), was used to amplify the expression cassettes containing *TyrH*$_{W13L, W369L}$ encoding tyrosine hydroxylase mutant, and *DODC* encoding DOPA decarboxylase[15] (i.e., ScTDH3p-*TyrH*$_{W13LW369L}$-ScTDH1t-ScCCW12p-*DODC*-ScADH1t). The gene encoding a norcoclaurine synthase (NCS) variant from *Coptis japonica* with the first 24 amino acids deleted[13] was synthesized by GenScript (Piscataway, NT). The ScTEF1 promoter and ScADH1 terminator were PCR amplified using the genomic DNA of BY4741 as templates. The ScTDH3p-*TyrH*$_{W13LW369L}$-ScTDH1t-ScCCW12p-*DODC*-ScADH1t cassette and the ScTEF1p-*CjNCS*-ScADH1t cassette were assembled to pRS413-*AnQut1co*, pRS415, and pRS416-*AnQut1co* linearized by NotI and SmaI using Gibson Assembly Cloning Kit (New England Biolabs, Ipswich, MA). All the recombinant vectors were confirmed by Sanger sequencing at the DNA facility (ISU, Ames, Iowa).

### High cell density fermentation

Recombinant *S. stipitis* strains carrying individual putative xylose-specific transporter and/or the refactored xylose pathway were freshly grown on SC-ura plates for 3–4 days. Single colonies were inoculated into 250 ml baffled flasks (VWR, Chicago, IL) containing 50 ml SC-ura plus 70 g/l glucose and 40 g/l xylose and grown for 12–24 h for seed

culture preparation. The resulting cultures were then transferred to 250 ml baffled flasks containing 50 ml fresh SC-ura plus 70 g/l glucose and 40 g/l xylose with an initial $OD_{600nm}$ at 10. Samples were taken every 24 h. Cell biomass was measured at $OD_{600nm}$, and the supernatant was stored in −20 °C freezer for sugar quantification. Glucose and xylose concentrations were quantified by HPLC. The high sugar concentrations rendered sufficient time to examine the uptake of xylose in the presence of glucose.

## Ethanol and shikimate production in *S. stipitis*

Recombinant *S. stipitis* strains were freshly grown on selective plates, and single colonies were inoculated into 250 ml baffled flasks (VWR, Chicago, IL) containing 20 ml SC-ura or SC-leu-ura plus 20 g/l glucose and 20 g/l xylose and grown for 12 h for seed culture preparation. For ethanol production, seed cultures were transferred into 100 ml serum bottles containing 60 ml fresh SC-ura plus 28 g/l glucose and 12 g/l xylose with 200 rpm orbital shaking. For shikimate production, seed cultures were transferred into 250 ml baffled flasks containing 20 ml fresh SC-leu-ura plus glucose/xylose mixture at different concentrations with 250 rpm orbital shaking. Initial cell $OD_{600nm}$ was ~0.2. Samples were taken every 24 h. Cell biomass was measured at $OD_{600nm}$, and the supernatant was stored in −20 °C freezer for sugar and metabolite quantification.

## Screening and characterization of shikimate transporters

Plasmids harboring the genes encoding putative shikimate transporters were transformed into BY4741, and transformants were grown on SC-his plates until colonies appeared. Three independent colonies of each transformation were first grown in SC-his containing 20 g/l glucose until saturation followed by 100-time dilution into glass tubes (Fisher Scientific, Waltham, WA) containing 5 ml SC-his plus 40 g/l glucose and 1 g/l shikimate. After 72 h, samples were taken and stored at −20 °C for shikimate usage assay.

Plasmids containing the codon-optimized shikimate transporters, AnQut1co and AnQut2co, were transformed into BY4741, and transformants were grown on SC-his plates until colonies appeared. Three independent colonies were first grown in SC-his containing 20 g/l glucose until saturation, and then transferred into glass tubes (Fisher Scientific, Waltham, WA) containing 5 ml SC-his plus 80 g/l glucose and 1 g/l shikimate with initial cell $OD_{600nm}$ of ~1 or 10. Samples were taken every 24 h for cell density measurement and shikimate usage assay.

## (*S*)-Norcoclaurine production

For monocultures, single colonies of NC1 were grown in SC-leu-his-ura media until saturation. Seed cultures were then transferred into fresh SC-his-leu-ura media plus 60 g/l glucose, 5 g/l ascorbic acid and 1 g/l SA with initial cell density of ~0.2. Samples were taken every 24 h for cell biomass and metabolite assays.

For co-cultures, single colonies of NC variant and SA4 were inoculated into SC-his-leu-ura medium and grown overnight. Three inoculation strategies were implemented in total, including (1) Simultaneous inoculation strategy I: the two seed cultures were transferred to fresh 2xSC-his-leu-ura media plus 60 g/l glucose, 20 g/l xylose, and 5 g/l ascorbic acid with a combined initial cell $OD_{600nm}$ of ~0.2, for which a series of cell ratio (e.g., Ss:Sc = 9:1; 3:1; 1:1; 1:3; 1:9) were tested. (2) Simultaneous inoculation strategy II: seed cultures were transferred to fresh 2xSC-his-leu-ura medium plus 30 g/l glucose, 30 g/l xylose, and 5 g/l ascorbic acid with the initial $OD_{600nm}$ of SA4 fixed at ~0.2 and the $OD_{600nm}$ of the *S. cerevisiae* variant varying at 0.2, 1, 3, 5, 7, 9, and 12. (3) Sequential inoculation strategy: the seed culture of SA4 was first transferred to a series of 2xSC-his-leu-ura medium, all containing 30 g/l glucose, 30 g/l xylose, and 5 g/l ascorbic acid with an initial cell density of ~0.2. NC1 culture was individually inoculated with a cell density of three to the corresponding SA4 culture that had been pre-cultivated

for a period ranging from 0 to 48 h (i.e., 0, 3, 6, 9, 12, 24, 36, and 48 h). Samples were taken every 24 h for cell biomass and metabolite assays.

## Analytical methods

Shikimate was detected by Waters HPLC system (Waters, Milford, MA) or by a colorimetric method as described elsewhere[23,78]. With respect to the colorimetric method, 100 µl diluted samples were first oxidized by Solution 1 containing 0.5% periodate and 0.5% sodium *m*-periodate followed by 45 min incubation at 37 °C. Then, the mixture was quenched by 100 µl Solution B (3:2 1 M NaOH/56 mM $Na_2SO_3$ (v/v)), and the absorbance at 382 nm was measured by a Synergy HTX multimode reader (Biotek, Winooski, VT). Shikimate, ethanol, acetate, and glycerol were detected using HPLC[23]. Briefly, 200 µl supernatant was collected and diluted 5–10 times with water. All samples were filtered through a 0.22-µm syringe filter (Simsii, Irvine, CA) prior to detection. Waters HPLC system was equipped with a binary HPLC pump, a 717 plus auto sampler, a column heater module, a 2998 photodiode array detector, and an Aminex HPX-87H column (300 × 7.8 mm) (Bio-Rad, Hercules, CA). The HPLC program was as follows: flow rate, 0.3 ml/min; column temperature, 30 °C; sample size, 10 µl; mobile phase, 5 mM sulfuric acid; and running time, 60 min/sample. PDA data extracted at 210 nm was compared to a standard curve made from commercially available shikimate (Sigma-Aldrich, St. Louis, MO). Refractive index data at 410 nm were compared to standard curves made from commercially available ethanol, acetate, and glycerol (Sigma-Aldrich, Saint Louis, MO).

Dopamine samples spiked with 100 µg/l dopamine-d4 (Millipore Sigma, Burlington, MA) as an internal standard for quantification. (*S*)-norcoclaurine and dopamine were analyzed by a 1290 Infinity Binary Pump UHPLC instrument equipped with a 6540 UHD Accurate-Mass Q-TOF mass spectrometer (Agilent Technologies) for fragmentation and mass detection. A volume of 7.5 µl of each sample was injected into the system. Samples were separated by UHPLC equipped with an Eclipse C18 1.8 µm 2.1 mm × 100 mm analytical column (Agilent Technologies) at 40 °C with a flow rate of 0.4 ml/min. Running solvent A was water with 0.1% formic acid, and solvent B was acetonitrile with 0.1% formic acid. All solvents were LC-MS grade (Fisher Chemical, Waltham, MA). The solvent conditions are as follows: 0% solvent B from 0 to 0.3 min, 0–5% solvent B from 0.3 to 10 min in a linear gradient, 5–100% solvent B from 10 to 13 min in a linear gradient, 100–0% solvent B from 13 to 15 min in a linear gradient, and 0% solvent B from 15 to 18 min. The system was run in a positive electrospray (ESI+) mode with a voltage of 200 V for fragmentation. Three targeted MS/MS selections were acquired: dopamine, *m/z* 154.1 with a narrow isolation width and a collision energy of 20 eV; the internal standard dopamine-d4, *m/z* 158.1 with a narrow isolation width and a collision energy of 20 eV; (*S*)-norcoclaurine, *m/z* 271.1 with a narrow isolation width and a collision energy of 23 eV. Data evaluation and peak quantitation were performed using Agilent MassHunter Qualitative Analysis software (version 10.0) (Agilent Technologies, Santa Clara, CA). Target peaks were quantified using the extracted LC-MS/MS ion chromatographs: dopamine 154.1 → 137.0593 ± 50 ppm, (*S*)-norcoclaurine 272.1 → 107.0498 ± 50 ppm, and the internal standard 158.1 → 141.0849 ± 50 ppm. Quantification of (*S*)-norcoclaurine was determined by a (*S*)-norcoclaurine calibration curve ranging from 25 to 800 µg/l, and dopamine was quantified by relative abundance to the internal standard and a standard curve ranging from 25 to 800 µg/l. The (*S*)-norcoclaurine and dopamine standards at different concentrations were prepared using spent medium collected from the strain Sc-pRS356 harboring empty plasmids. All samples were filtered through 0.22-µm syringe filters (Simsii, Irvine, CA) prior to detection. (*S*)-norcoclaurine and dopamine standards were purchased from Sigma-Aldrich (Saint Louis, MO).

## Statistical analysis and reproducibility

All data analysis was performed using Prism software (GraphPad, San Diego, CA). Independent unpaired *t*-test was used for two-sided comparison. Differences were considered statistically significant when *p* value is <0.05. All experiments were conducted in biological triplicates, quadruplicates, or quintuplicates.

## Reporting summary

Further information on research design is available in the Nature Portfolio Reporting Summary linked to this article.

## Data availability

All data supporting the findings in this study are available within the Article and Supplementary Information files. All the plasmids containing the refactored xylose pathway, the shikimate pathway, xylose transporters, shikimate transporters, and the norcoclaurine pathway that support the findings of this study are available from the corresponding author upon request. Source data are provided with this paper.

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

## Acknowledgements

This work is supported by the National Science Foundation Grants (1716837, EEC-0813570, and 1749782), the Iowa State University

Biobased Chemical/Bioproduct Seed Grant (PG110304), and the Manley Hoppe Professorship held by Professor Jacqueline Shanks. We thank Dr. Thomas W. Jeffries (Emeritus Professor of Bacteriology at the University of Wisconsin-Madison and Founder of Xylome Corporation) for sharing the strain *S. stipitis* FLP-UC7 (ura3-3, NRRL Y21448) and Dr. John E. Dueber for sharing the plasmid p2249. We thank Dr. Shawn M. Rigby for flow cytometry analysis. We acknowledge the W.M. Keck Metabolomics Research Laboratory (Office of Biotechnology, Iowa State University, Ames IA) for providing analytical instrumentation and thank Dr. Lucas J. Showman and Dr. Ann M. Perera for their assistance and support for the quantification of (*S*)-norcoclaurine, dopamine, and 4-hydroxyphenylacetate.

## Author contributions

Z.S. supervised the research and edited the manuscript. M.G. and Y.Z. designed the research, conducted experiments, analyzed data, and wrote and edited the manuscript. Z.Y. contributed to assess the impact of individual genes in relieving CCR. Q.S. contributed to the HCDF sugar utilization assay. P.V.B. performed the SA production assay.

## Competing interests

Z.S. and M.G. are listed as inventors on the patent "Methods and Compositions for Production of Aromatic and Other Compounds in Yeast" (U.S. Patent No. 10,787,672). Z.S. is the founder of the startup company, ESTose Biorenewables LLC, Ames, IA. The remaining authors declare no competing interests.
