## [Peer Review File · Nature Communications]

Novel Xylose and Shikimate Transporters Facilitates Microbial Consortium as a Chassis for Benzylisoquinoline Alkaloid ProductionReviewers' Comments:

Reviewer #1:

Remarks to the Author:

The submitted manuscript reports a novel approach for biosynthesis of a valuable natural product S-norcoclaurine. A microbe, yeast *S. stipitis*, was first genetically modified to enable effective utilization of glucose and xylose for producing shikimate. Another yeast, *S. cerevisiae*, was subsequently engineered to convert shikimate to the final product through functional expression of desired pathway enzymes. The co-cultivation of these two species allowed for the streamlined production of S-norcoclaurine from renewable substrates glucose and xylose. Overall, this is a novel and well-organized study with a clear theme and valuable findings. There are a few issues that need to be addressed before publication on Nature Communications.

Specific comments:

The use of the engineered consortium is one of the key highlights of this work. To this end, it would be great interest to ascertain the consortium population composition change with time, i.e., the variation of the consortium members during the co-cultivation process. This can be done by using the phenotypic differences between the two species. The results can indicate how the consortium members collaborate, in a dynamically manner, with each other to strengthen the desired product biosynthesis. If there are technical challenges for achieving this in the context of the consortium, the authors are suggested to make appropriate discussion on this issue.

It was found that using shikimate as the connecting molecule presents a few technical challenges (including shikimate importation barrier, cell inhibition by shikimate kinase expression). Would it be helpful to use other molecules, such as tyrosine, as the connecting molecule? On the other hand, for high-value products, the use of inexpensive substrates is not a top concern. In that sense, can tyrosine be directly used as the starting molecule for S-norcoclaurine biosynthesis (i.e., using a mono-culture of *Sc* fed with tyrosine or shikimate)? Please discuss further about this issue to better justify the motivation of the consortium-based strategy.

It is an interesting finding that delayed addition of the downstream strain to the consortium changed the production performance. Apparently, too early or late initiation of the co-culturing process changed the interaction dynamics between the consortium members. Simultaneous inoculation may start overly early competition for resources (e.g., glucose), whereas late co-culturing could leave insufficient resources for the downstream species to do its part. Are there more data, especially the time profile data, for the consortium growth and biosynthesis behavior under different delayed inoculation strategies? Can the author provide more discussion about this?

What is the shikimate accumulation profile under different inoculation ratios? This is an essential factor for the developed consortium system and should be carefully examined. Also, did the author make attempts to investigate the accumulation of other key pathway intermediates, such as 4-HPAA and dopamine? These are likely limiting molecules for the overall biosynthesis process.

Along with the comment above, to promote shikimate assimilation and utilization by the downstream *Sc* strain, have the authors attempted to increase the transporter's copy number? This may be particularly interesting when there is still shikimate accumulation in the consortium system, as shown in Fig. S6.

Fig 3a. It would be useful to show the cell density change and align it with the carbon substrate consumption profile. Also, it is interesting to note that, after the depletion of glucose and xylose, the engineered strain with SpXut1 was able to consume ethanol. This may provide additional benefits for engineering microbial consortia; more discussion about this finding can help expand the potential utility of the developed strategy.

Fig 4c shows the shikimate profiles of the cell culture inoculated at different ODs. Specifically, it appears that the use of low and high ODs does not make much difference in the shikimate utilization efficiency. However, the corresponding explanation and discussion are not present in the main text.

Unlike same-species consortia, the consortium of this work used two yeast species *S. stipitis* and *S. cerevisiae* with different growth rates, which can result in imbalance of the biosynthesis capabilities during the co-cultivation process. Although this can be offset to some degree by the inoculation ratio optimization, this issue can be magnified at large-scale cultivation and production. Can the authors discuss about the challenges and potential solution for scale-up of the developed consortium system?

The confirmation of the expected product's chemical identity is critical for this work. Please provide the MS/MS spectra data for (S)-Norcochlorine standard and the one produced by engineered co-culture.

For xylose pathway refactoring, among the five genes that were over-expressed to improve the xylose utilization, are there any data to suggest which one(s) contribute more than others?

Line 221 "The activation at the transcriptional level through promoter swapping confirms that SsXut1, SsXut2, and SsXut4 are all functional xylose-specific transporters free from glucose inhibition." It appears that the findings here showed that the xylose transporters with an engineered promoter were effective for improving the xylose uptake, and thus they are confirmed to be effective transporters. However, it is a bit misleading to state that "these transporters are free from glucose inhibition", as it cannot be ruled out that the native promoters for these promoters are still regulated by glucose. The sentence here may need to be rephrased to make the point clear.

For triplicate samples used for the experiments, are they biological or technical replicates? Please clarify in the figure caption or the method section.

There are some typos that need to be addressed. For example, line 63, "imposeing" should be "imposing".

Reviewer #2:
Remarks to the Author:
Overview

This article describes transporter discovery, first on xylose transporters, which has a long history, and then on shikimate transporters, which is novel. While the majority of transporter work has historically focused on biomass utilization, the authors here re-envision how transporters may be used to optimize a consortia. This is not for biomass utilization, but for providing nutrients to maintain two separate species and to rewire the metabolic flux of a consortium member so that shikimate is overproduced. They then go on to produce alkaloids with a co-culture. These products are particularly of interest. Overall, this is a compelling narrative that involves non-obvious insights into improving titers of consortia, highlighting transport as a key step for nutrient import (not novel) and for intermediate exchange (novel). However, at times the narrative loses a cohesive thread and can read like three different impactful findings (SpXUT1 story, shikimate transporter story, consortia story). The article could chiefly benefit from a revision to streamline and focus the elements of the paper to maximize impact to the readership of Nature Communications.

I recommend acceptance after minor revision on the following points:

- Engineering co-consumption of glucose and xylose on the scale that the authors report is of interest alone, it's almost a shame to include it with the other impactful results, but each element of the manuscript as written has impact.

- If both *S. stipitis* and *S. cerevisiae* suffer from CCR, why not make a CCR optimized *S. cerevisiae* strain that produces norclaurine? The shikimate transport issue wouldn't then be a problem. The only answer I could find to this question was late in the paper, line 444, and in conclusion line 482.
- Line 162 – ameliorate the availability of E4P. I don't understand this, do the authors mean alleviate an E4P bottleneck? Increase E4P pools? I'm not sure.
- Line 165 – please cite a review or reviews that highlight the ubiquity of this problem.
- Line 173 – Why not a hydrolysate? Yeast fermentation media are pretty complex mixtures, I'm not an expert on purification but it doesn't seem like a mixture of glucose, xylose, lignin derivatives would be that troublesome unless there were huge amounts of furfurals and aromatics. It is plant material after all and plants are often sourced for nutraceuticals and pharmaceuticals. It seems the authors are responding to a past criticism here but I'm not sure they need to eliminate this possibility so strongly, although a purified G-X mixture is obviously preferable to a complex hydrolysate.
- Line 193 – other bottlenecks exist. This seems like a leap of a claim. Is this only because alleviating CCR on the PPP didn't completely remove the inhibition?
- Line 227 – serving as repositories. I would revise this to “reservoirs” or “sources”
- The first few lines of the Discussion are confusing as they fluidly transition from *S. cerevisiae* to *S. stipitis* without a clear difference.
- Line 408 and 409 – xylose transport has been known as a bottleneck for at least twenty years, and Kotter, Zhao, Alper, de Waal, and Boles, among others, have done a lot of work (uncited here save for Zhao I think) to engineer transport. I think it is a little bit of an overstatement to say that it “emerged as a bottleneck”
- Line 412 – I think the authors need to be more specific about what their study proves about transport and metabolic engineering because, to my previous comment, I think this has been proved many times in the literature.
- Line 414 – SA is used instead of shikimate as an unexplained abbreviation
- Line 422 – *Candida albicans*
- Line 422 – this is the first mention of Ca and Dh transporters being investigated – shouldn't all transporters be mentioned in the results?
- Line 462 – “all that ever reported” - please revise this
- Line 469 – I'm not sure how the paper leads to this statement. The statement seems completely separate from the point of the paper, which is transport and consortium engineering.

Reviewer #3:

Remarks to the Author:

The manuscript reports a consortium platform consisting of two different yeast strains to produce a target molecule. The yeast consortium demonstrated effective partitioning of metabolic pathways into different yeast strains to produce an aromatic amino acid derivative. In order to implement the yeast consortium, the authors identified a putative xylose-specific transporter for *Schefferomyces stipitis* and quinate permeases for *Saccharomyces cerevisiae* to enable enhanced production of (S)-norcoclaurine from a mixture of glucose and xylose. The presented results in the manuscript are promising, but additional experiments and revisions of the manuscript might be necessary.

Specific comments:

1. Line 188-192, Please define HCDF – high cell density fermentation?
2. According to S1 figure, *S. stipitis* consumed glucose efficiently even with low initial OD inoculations under YP conditions. However, in Fig 2a, it took 5 days for both control and mutant strains to consume 20-30 g/L glucose even with high-density inoculations under SC-ura conditions. These are extremely slow glucose consumption rates. Does the 20kb PMG xyl plasmid has negative effects on yeast growth?
3. Line 222-223, the expression of SsXUT1, 2, and 4 indeed enhanced co-utilization of glucose and

xylose. If Fig 2A and B are compared, it seems that the expression of those transporters improved not only xylose uptake but also glucose consumption. Are they xylose-specific transporters? The statement that SsXUT1,2 and 4 are xylose-specific transporters without glucose inhibition might not be correct. Further experimental validation such as glucose/xylose uptake kinetics with and without glucose will need to be conducted. Those transporters still might exhibit glucose transport activities.

4. For Fig 2B, this reviewer strongly recommends providing the sugar consumption and growth profiles with the best transporter candidates. It will visualize the effects of co-consumption.

5. Previous studies on transporter engineering by Young et al. and Farwick et al. have identified critical amino acid residues responsible for co-transport or/and xylose-specific transport. Have the authors identified any of those critical residues in Ss or SpXUTs? The comparison result can be discussed in the Discussion.

6. Line 238-239 – Hxt11 mutant transporter has been reported to be a co-transporter of glucose and xylose in a hxt-null background with reduced kinetic properties. Given that Ss strain has native xylose transporters intact, the statement, “neither transporter conferred xylose uptake in *S. stipitis*” does not make sense.

7. Line 241-244, while fermentation data showed enhanced xylose uptake in the presence of glucose, further sugar uptake kinetics experiment might be necessary to support the statement “xylose-specific transporter”

8. Line 350-352 – the observed that *S. cerevisiae* overexpressing Aro1 or AroL with Aro7mut exhibited growth defects when shikimate is present. Why previously constructed strains with the addition of the upstream pathways (i.e., shikimate overproducing) did not show this toxicity? Maybe intracellular metabolomics study can be conducted to clarify this.

9. In Fig 5b, the authors aimed to demonstrate the beneficial effects of sequential inoculation of the strains on S-norcochlorogenic acid production. However, after initial inoculation of the NC3 strain into SA4 grown media, S-norcochlorogenic acid concentration did not change within the next 96 hours (even decreased to some extent till 72 hr). It is quite difficult to understand Fig 5b. It would be helpful to provide a full fermentation profile showing sugar consumption, shikimate, S-norcochlorogenic acid production, and other metabolites (ethanol, glycerol, and acetate) in the best condition that yielded 4.2 mg/L.

10. Liu et al. Rewiring carbon metabolism in yeast for high-level production of aromatic citraconic acid appear twice in reference and main text (41, 50).

Reviewer #1 (Remarks to the Author):

The submitted manuscript reports a novel approach for biosynthesis of a valuable natural product S-norcoclaurine. A microbe, yeast *S. stipitis*, was first genetically modified to enable effective utilization of glucose and xylose for producing shikimate. Another yeast, *S. cerevisiae*, was subsequently engineered to convert shikimate to the final product through functional expression of desired pathway enzymes. The co-cultivation of these two species allowed for the streamlined production of S-norcoclaurine from renewable substrates glucose and xylose. Overall, this is a novel and well-organized study with a clear theme and valuable findings. There are a few issues that need to be addressed before publication on Nature Communications.

Response: Thank you for all the positive comments. Below please see our responses to individual questions. The texts highlighted in yellow have been added to the revised manuscript.

Specific comments:

1. The use of the engineered consortium is one of the key highlights of this work. To this end, it would be great interest to ascertain the consortium population composition change with time, i.e., the variation of the consortium members during the co-cultivation process. This can be done by using the phenotypic differences between the two species. The results can indicate how the consortium members collaborate, in a dynamically manner, with each other to strengthen the desired product biosynthesis. If there are technical challenges for achieving this in the context of the consortium, the authors are suggested to make appropriate discussion on this issue.

Response: Indeed, we attempted to measure changes in the population composition of a consortium over time by leveraging phenotypic differences between the two species. Specifically, *S. stipitis* can naturally assimilate xylose, while *S. cerevisiae* can only utilize glucose. To determine the population ratio of these two consortium members during fermentation, we conducted a 120-h co-culture fermentation using the best-performing upstream and downstream strains (named SA4 and NC1) under the optimal consortium condition. We collected fermentation samples every 24 h, diluted them 10-1000 folds, and plated them on three different agar plates that allowed only SA4 (SC-Leu-Ura + 2% xylose) or both SA4 and NC1 (SC-His-Leu-Ura + 2% glucose and YPA + 2% glucose) to survive. However, the numbers of colonies appearing on the YPA+2% glucose plates were 2.6-13.6-fold higher than those on the SC-His-Leu-Ura+2% glucose plates, indicating an issue of plasmid instability in SA4 or NC1 during a five-day fermentation. As a result, we were unable to accurately ascertain the ratio of consortium members over time. To address these challenges, we have added corresponding discussion in the manuscript to propose some feasible strategies.

(Line 517-528) In parallel, we also attempted to measure the population compositions of the SA4/NC1 consortium using their phenotypic difference that only *S. stipitis* enables the catabolism of xylose. We collected consortium samples every 24 h, diluted 10-1000 folds, and plated them on three different agar plates that allowed only SA4 (SC-Leu-Ura + 2% xylose) or both SA4 and NC1 (SC-His-Leu-Ura + 2% glucose and YPA + 2% glucose) to survive. However, we were unable to accurately ascertain the ratio of the consortium members over time presumably due to plasmid instability. The numbers of colonies appearing on the YPA + 2% glucose plates were 2.6-13.6-fold higher than those on the SC-His-Leu-Ura + 2% glucose plates. To address this challenge, genomic integration of the pathway needs be performed in the future to achieve higher genetic stability. Additionally, accurately monitoring population dynamics in real time can be achieved using multi-color flow cytometry analysis(1, 2) that

involves quantifying different populations with corresponding integrated fluorescent proteins-encoding genes.

2. It was found that using shikimate as the connecting molecule presents a few technical challenges (including shikimate importation barrier, cell inhibition by shikimate kinase expression). Would it be helpful to use other molecules, such as tyrosine, as the connecting molecule? On the other hand, for high-value products, the use of inexpensive substrates is not a top concern. In that sense, can tyrosine be directly used as the starting molecule for *S*-norcoclaurine biosynthesis (i.e., using a mono-culture of *Sc* fed with tyrosine or shikimate)? Please discuss further about this issue to better justify the motivation of the consortium-based strategy.

Response: There are two questions in this comment:

(1) (Line 457-465) At the initial stage of the project, tyrosine was also being considered as a potential connecting molecule. We did not select it further due to the initial concern of the potential competition with protein synthesis and later found that the low solubility of tyrosine in water (0.5 g/L) at room temperature and neutral pH(3) presents a practical challenge in the pre-test experiments related to supplementation of a high-concentration precursor. Alternative methods, such as separately feeding a high-concentration tyrosine stock solution prepared in 1N HCl into the fermentation medium, have been attempted. Unfortunately, this approach has resulted in the inhibition of cell growth due to low pH, leading to low (*S*)-norcoclaurine productions (Figure S14). Therefore, selection of shikimate as the connecting molecule for (*S*)-norcoclaurine production has the advantage due to its high solubility in water.

Figure S14. (*S*)-Norcoclaurine production (a) and cell growth profiles (b) of the strain *Sc*-TDC fed with various concentrations of tyrosine in 72 h. The recombinant *S. cerevisiae* strain *Sc*-TDC contains the plasmid pRS415-Tyr^{H_{W13LW369L}}-DODC-C_jNCS. The initial OD_{600nm} was set at ~0.2. The fermentation medium SC-leu was supplemented with different concentrations of tyrosine and 5 g/L L-ascorbic acid. Tyrosine stock at 50 g/L were prepared in 1N HCl. Samples were collected every 24 h. Error bars represent the standard deviations of biological triplicates.

(2) We already attempted direct supplementation using 1 g/L of shikimate (Figure S7) and different cell densities of the NC1 monoculture. The highest production of (*S*)-norcoclaurine (initial OD₆₀₀ at 5) was about 3 mg/L, which was less than 1/3 of the titer enabled by the co-culture of NC1 and SA4 under a very similar fermentation condition. Thus, co-culturing *Ss* and *Sc* variants increased the titer of (*S*)-norcoclaurine through continuous assimilation of shikimate and production in real-time instead of one-time feeding.

Figure S7. (*S*)-norcoclaurine production and shikimate utilization by NC1 monocultures. (a) The titers of (*S*)-norcoclaurine yielded by different initial cell densities. (b) Shikimate utilization of the strain NC1 with different initial cell densities. NC1 was inoculated at different cell densities in 2xSC-his-leu-ura supplemented with 80 g/L glucose and 1 g/L shikimate. Samples were taken after four-day fermentation. Error bars represent the standard deviations of biological triplicates.

3. It is an interesting finding that delayed addition of the downstream strain to the consortium changed the production performance. Apparently, too early or late initiation of the co-culturing process changed the interaction dynamics between the consortium members. Simultaneous inoculation may start overly early competition for resources (e.g., glucose), whereas late co-culturing could leave insufficient resources for the downstream species to do its part. Are there more data, especially the time profile data, for the consortium growth and biosynthesis behavior under different delayed inoculation strategies? Can the author provide more discussion about this?

Response: We tested the impact of the mixing time of two species on (*S*)-norcoclaurine production using the consortium of SA4/NC1 and collected samples at 96 h of fermentation. To address the question, we added the following texts in the manuscript.

(Line 383-394) In addition to the inoculated amount of the strain NC1, another variable is the growth stage of the strain SA4. Considering the potential benefit of allowing SA4 to grow first and accumulate shikimate to a decent level prior to introducing NC1, we tested a sequential inoculation strategy. Specifically, NC1 was inoculated at a cell density of three into the SA4 culture that had been cultivated for a varying period ranging from 0 h to 48 h. The result showed that delaying the addition of NC1 resulted in drastic decreases in the (*S*)-norcoclaurine titer from 16.1 mg/L (at 0 h) to 0.32 mg/L (at 48 h) (Figure 5b), suggesting that early inoculation of NC1 was more effective in achieving (*S*)-norcoclaurine production. The delayed inoculation of NC1 led to an increased xylose consumption and an undesired shikimate buildup by SA4, particularly if SA4 was pre-cultured for 36 h and 48 h. Consequently, there were insufficient nutrients in the medium available for NC1, which significantly reduced (*S*)-norcoclaurine titer (Figure 5b and Figure S10).

Figure 5. Optimizing (S)-norcoclaurine production by yeast consortia.(b) (S)-norcoclaurine titers of the consortia with a sequential inoculation strategy. *S. stipitis* SA4 was first grown in fermentation medium (2x SC medium containing 30 g/L glucose, 30 g/L xylose, and 5 g/L L-ascorbic acid). After 0 h, 3 h, 6 h, 9 h, 12 h, 24 h, 36 h and 48 h, *S. cerevisiae* NC1 was introduced with an OD_{600nm} of three into the corresponding *S. stipitis* SA4 culture. Samples were collected after 96 h of fermentation.

Figure S10. Quantification of cell density, accumulation of dopamine and shikimate, and residual glucose and xylose in the cocultures consisting of SA4 and NC1 with a sequential inoculation strategy. *S. stipitis* SA4 was first grown in fermentation medium (2xSC medium containing 30 g/L glucose, 30 g/L xylose, and 5 g/L L-ascorbic acid) with an initial cell density of 0.2-0.3. After 0 h, 3 h, 6 h, 9 h, 12 h, 24 h, 36 h, and 48 h, *S. cerevisiae* NC1 was introduced into the corresponding *S. stipitis* SA4 culture with an OD_{600nm} of three. Samples were collected after 96 h of fermentation. (a) Cell densities of the cocultures were measured at 600 nm after 96 h of fermentation. (b) Dopamine accumulation and residual shikimate, glucose, and xylose were monitored in the cocultures. Error bars represent the standard deviations of biological triplicates.

4. What is the shikimate accumulation profile under different inoculation ratios? This is an essential factor for the developed consortium system and should be carefully examined. Also, did the author make attempts to investigate the accumulation of other key pathway intermediates, such as 4-HPAA and dopamine? These are likely limiting molecules for the overall biosynthesis process.

Response: We split this comment into two questions:

(1). We provided the accumulation profiles of shikimate and dopamine (also the remaining sugars) under different inoculation ratios in Figure S9 and added the following texts.

(Line 364-368) We also quantified cell density, accumulation of dopamine and shikimate, and residual glucose and xylose in the cocultures (Figure S9). Consistent with the highest titer yielded when NC1 was inoculated with an initial OD_{600nm} of 3, an ignorable level of shikimate was detected. Lower amounts of inoculated NC1 resulted in high levels of shikimate accumulation whereas higher initial amounts of NC1 resulted in undesired buildups of xylose.

Figure S9. Quantification of cell density, accumulation of dopamine and shikimate, and residual glucose and xylose in the cocultures consisting of SA4 and NC1 with various initial Ss:Sc inoculation ratios. The initial cell density of SA4 was 0.2-0.3 whereas the OD_{600nm} of NC1 was adjusted at different levels. The two strains were simultaneously introduced. Samples were collected after 96 h of fermentation. (a) Cell densities of the cocultures were measured at 600 nm after 96 h of fermentation. (b) Dopamine accumulation and residual shikimate, glucose, and xylose were monitored in the cocultures. Error bars represent the standard deviations of biological triplicates.

(2). (S)-Norcoclaurine is synthesized *via* the condensation of dopamine and 4-HPAA. In yeasts, 4-HPAA is produced endogenously by the native Ehrlich pathway and can be rapidly transformed to the corresponding fusel acid 4-hydroxyphenylacetic acid (4-HPAC) or alcohol tyrosol there(4). This makes it challenging to accurately quantify the concentration of 4-HPAA. Instead, we measured the accumulation of dopamine and added the following texts:

(Line 402-408) In addition, more than three-fold higher accumulation of dopamine was observed during the coculture. This is consistent with the previous monoculture studies in *S. cerevisiae*(5) and *S. stipitis*(6) that reported dopamine accumulation, suggesting that either low NCS expression level or insufficient 4-HPAA precursor supply limited (S)-norcoclaurine formation. The native Ehrlich pathway could also play a role as it might rapidly transform 4-HPAA to the corresponding fusel acid, 4-hydroxyphenylacetic acid (4-HPAC) or alcohol tyrosol(4).

Figure 5. Optimizing (S)-norcoclaurine production by yeast consortia.....(c) The production time course of (S)-norcoclaurine and dopamine in the consortium under the optimal fermentation condition. The initial cell density of SA4 was 0.2-0.3 and NC1 was simultaneously introduced into the culture with an OD_{600nm} of three. Samples were collected every 24 h post the starting of the co-culture. Error bars represent the standard deviations of biological quadruplicates.

5. Along with the comment above, to promote shikimate assimilation and utilization by the downstream Sc strain, have the authors attempted to increase the transporter's copy number? This may be particularly interesting when there is still shikimate accumulation in the consortium system, as shown in Fig. S6.

Response: We would like to thank the reviewer for asking this interesting question. Following the reviewer's comment, we tested the effect of multi-copy number of shikimate transporter on shikimate import and (S)-norcoclaurine production. The following text and figures were added to the Discussion section.

(Line 501-516) Additionally, considering the best inoculation strategy so far still led to an accumulation of shikimate at 0.41 g/L after 120 h of fermentation (Figure S12), we sought to further increase (S)-norcoclaurine production by increasing the copy number of shikimate transporter gene AnQut1co. Two more strains consisting of two-copy and three-copy AnQut1co were constructed by incorporating AnQut1co into pRS415-TyrH*-DODC-NCS (named NC2) and then into pRS416-TyrH*-DODC-NCS (named NC3) (Table S2). A strain named Sc-pRS356 containing empty plasmids pRS413, pRS415, and pRS416 was included as a negative control. Unexpectedly, doubling and tripling the dose of AnQut1co significantly decreased the (S)-norcoclaurine titer from 5.7 mg/L (NC1) to 1.9 mg/L (NC2) and 1.3 mg/L (NC3) (Figure S15). Correspondingly, decreases in cell density and higher contents of residual shikimate were observed in the cocultures of SA4/NC2 and SA4/NC3 (Figure S16). This result might be correlated with the previous observation of significantly reduced growth rates of the strains overexpressing shikimate kinase (i.e., EcAroL or ScAro1) (Figure 4a and Figure S6). To study the causality, future efforts should be first focused on investigating the ATP drainage in these variants.

Figure S15. (S)-norcochlorine production by yeast consortia with different copies of AnQut1co. After the strain SA4 was inoculated into fermentation medium with a starting OD_{600nm} of 0.2-0.3 and cultivated for 12 h, NC variants (NC1, NC2, and NC3) with OD_{600nm} of three was inoculated into the culture. Error bars represent the standard deviations of biological triplicates. Asterisks indicate p values generated by independent unpaired t-tests. ** $p < 0.01$. Error bars represent the standard deviations of biological quintuplicates. Samples were collected every 24 h post the starting of co-culturing.

Figure S16. Comparison of co-cultures including one, two, and three-copy of AnQut1co. Quantification of (a) shikimate accumulation, (b) OD_{600nm} , (c), residual glucose, and (d) residual xylose in the co-cultures consisting of SA4 and different NC variants (NC1, NC2 and NC3). After the strain SA4 was inoculated into fermentation medium with a starting OD_{600} of 0.2-0.3 and cultivated for 12 h, NC variants and the control (Sc-pRS356) with OD_{600nm} of three was inoculated into the culture. Samples were collected every 24 h post the starting of co-culturing. Error bars represent the standard deviations of biological triplicates.

6. Fig 3a. It would be useful to show the cell density change and align it with the carbon substrate consumption profile. Also, it is interesting to note that, after the depletion of glucose and xylose, the engineered strain with SpXut1 was able to consume ethanol. This may provide additional benefits for engineering microbial consortia; more discussion about this finding can help expand the potential utility of the developed strategy.

Response: We have added the cell growth profile into Figure 3a. After the depletion of mixed sugars, the engineered strain *S. stipitis* strain (Ss-*xyl*-SpXUT1) with CCR relief started to consume ethanol for energy supply and biomass formation, resulting in another increase in cell density from 48 h-120 h.

Figure 3. The fermentation and shikimate production profiles after CCR relief. (a) The fermentation profile of *S. stipitis* strain (Ss-*xyl*-SpXUT1) with CCR relief. Initial cell density (OD_{600nm}) was set at ~10.

7. Fig 4c shows the shikimate profiles of the cell culture inoculated at different ODs. Specifically, it appears that the use of low and high ODs does not make much difference in the shikimate utilization efficiency. However, the corresponding explanation and discussion are not present in the main text.

Response: This is a good catch. We have added the following texts in the manuscript. Note that Figure 4c mentioned in the reviewer's comment was changed to Figure 4d in the revised manuscript.

(Line 315-320) The indistinction between LCDF and HCDF in terms of shikimate import could be attributed to the fact that the cell density of *S. cerevisiae* only takes ~140 minutes to double in the synthetic medium, which means that it takes only about three generations to reach an OD_{600nm} of 10 from an OD_{600nm} of 1. As a result, no significant difference was observed between LCDF and HCDF for shikimate import after sampling every 24 hours.

8. Unlike same-species consortia, the consortium of this work used two yeast species *S. stipitis* and *S. cerevisiae* with different growth rates, which can result in imbalance of the biosynthesis capabilities during the co-cultivation process. Although this can be offset to some degree by the inoculation ratio optimization, this issue can be magnified at large-scale cultivation and production. Can the authors discuss about the challenges and potential solution for scale-up of the developed consortium system?

Response: We have added the following texts into the Discussion section.

(Line 494-501) As the majority of the synthetic microbial consortia are complex with dynamic compositions, one of the key factors in their design is to prevent the dominance of one species over another due to a shorter doubling time or an advantage of utilizing certain substrates. To achieve a stable coexistence among the species, conventional strategies such as titration of the inoculum ratio and optimization of mixing time can be exploited. In our study, the Ss/Sc consortium yielded the highest (*S*)-norcochlorine titer (11.5 mg/L) when these two determinants

were optimized simultaneously (i.e., an initial OD_{600nm} of three for NC1 and 0.2-0.3 for SA4; simultaneous inoculation of the two seed cultures at 0 h).

(Line 530-541) Lastly, for industries to eventually utilize microbial consortia for chemical production, self-regulation is a key feature desired to be established in co-cultures because manual titration of cell densities of individual species will be challenging along with the scale-up processes. To this end, relieving the competition by using distinct carbon sources and establishing symbiotic relationships have been applied to improve the consortium stability and robustness(7-10). Recently, a multi-metabolite crossing feeding strategy was developed to strengthen the connection between species. Relying on amino acid anabolism, energy metabolism, and a caffeine-responsive biosensor, autonomous regulation of strain ratios was achieved in a three-strain co-culture for *de novo* biosynthesis of silybin and isosilybin(11). Such a strategy is highly valuable for designing synthetic consortia that involve distinct species with different natural habitats including various substrate spectrums, metabolic performances, and metabolite profiles(12). In all cases, reliable modeling tools to predict the interactions in microbial consortia that span different spatiotemporal scales combined with high-resolution multi-omics analysis are highly desired to harness the full potential of microbial consortia for large-scale industrial production(13).

9. The confirmation of the expected product's chemical identity is critical for this work. Please provide the MS/MS spectra data for (S)-Norcoclaurine standard and the one produced by engineered co-culture.

Response: We have added the requested LC-MS/MS analysis results of (S)-norcoclaurine and dopamine in Figure S11.

Figure S11. LC-MS/MS analysis of (a) (S)-norcoclaurine and (b) dopamine in the supernatant of the SA4/NC1 co-culture after 72 h fermentation. Fermentation was performed in 2×SC medium containing 30 g/L glucose, 30 g/L xylose, and 5 g/L L-ascorbic acid. Red boxes indicate the four signature [M+H]⁺ ions of (S)-norcoclaurine standard ([m/z] 272.1, 255.1, 161.0, and 107.0) and dopamine standard ([m/z] 154.1, 137.0, 119.0, and 91.0).

10. For xylose pathway refactoring, among the five genes that were over-expressed to improve the xylose utilization, are there any data to suggest which one(s) contribute more than others?

Response: We would like to thank the reviewer for asking this interesting question. Following the reviewer's comment, we have performed the corresponding experiments and added the following text into the manuscript.

(Line 190-199) To assess whether all six genes in the plasmid pMG-*xyl*-SpXUT1 are necessary for efficient mixed-sugar utilization, we constructed another six plasmids by removing one gene at a time from pMG-*xyl*-SpXUT1 followed by individual plasmid transformation into *S. stipitis*. In contrast to the control strain harboring the complete set of six genes, removal of any gene from the combination resulted in a drastically reduced efficiency of mixed-sugar utilization, with xylose assimilation being particularly affected (Figure S4). Additionally, each of the variants lacking an intact refactored xylose conversion pathway displayed a decrease in biomass formation, compared to both the control strain with the full set of six genes and the variant with only SpXut1 removed. This might be due to an attenuated cofactor balance (especially when Xyl1 and Xyl2 were dropped off) as well as the reoccurrence of CCR(14).

Figure S4. The sugar consumption (a) and cell growth (b) profiles of the *S. stipitis* strains cultured in SC-ura medium containing 70 g/L glucose and 40 g/L xylose. The derived variants were created by removing one gene at a time from the control strain *Ss-xyl*-SpXUT1 (i.e., the refactored xylose conversion pathway plus SpXut1 transporter). HCDF was conducted with an initial cell density of ~10. Samples were collected every 24 h. Error bars represent the standard deviations of biological triplicates.

11. Line 221 “The activation at the transcriptional level through promoter swapping confirms that SsXut1, SsXut2, and SsXut4 are all functional xylose-specific transporters free from glucose inhibition.” It appears that the findings here showed that the xylose transporters with an engineered promoter were effective for improving the xylose uptake, and thus they are confirmed to be effective transporters. However, it is a bit misleading to state that “these transporters are free from glucose inhibition”, as it cannot be ruled out that the native promoters for these promoters are still regulated by glucose. The sentence here may need to be rephrased to make the point clear.

Response: We have removed “free from glucose inhibition” from the manuscript (Line 169).

12. For triplicate samples used for the experiments, are they biological or technical replicates? Please clarify in the figure caption or the method section.

Response: All the replicates used for the experiments are biological replicates. We have added “All experiments were conducted in biological triplicates, quadruplicates, or quintuplicates” in the Methods section under “Statistical analysis and reproducibility (Line 748-749) and also indicated in the captions of figures.

13. There are some typos that need to be addressed. For example, line 63, “imposeing” should be “imposing”.

Response: We have corrected all the typos and grammatical mistakes in the manuscript.

Reviewer #2 (Remarks to the Author):

Overview

This article describes transporter discovery, first on xylose transporters, which has a long history, and then on shikimate transporters, which is novel. While the majority of transporter work has historically focused on biomass utilization, the authors here re-envision how transporters may be used to optimize a consortia. This is not for biomass utilization, but for providing nutrients to maintain two separate species and to rewire the metabolic flux of a consortium member so that shikimate is overproduced. They then go on to produce alkaloids with a co-culture. These products are particularly of interest. Overall, this is a compelling narrative that involves non-obvious insights into improving titers of consortia, highlighting transport as a key step for nutrient import (not novel) and for intermediate exchange (novel). However, at times the narrative loses a cohesive thread and can read like three different impactful findings (SpXUT1 story, shikimate transporter story, consortia story). The article could chiefly benefit from a revision to streamline and focus the elements of the paper to maximize impact to the readership of Nature Communications.

I recommend acceptance after minor revision on the following points:

1. Engineering co-consumption of glucose and xylose on the scale that the authors report is of interest alone, it’s almost a shame to include it with the other impactful results, but each element of the manuscript as written has impact.

Response: We would like to express our sincere gratitude to the reviewer for the positive comments. Following this comment, we have moved some texts and Figure 6a that were originally in the main texts to Supporting Information. On the other hand, some additional

discussions were added as responses to the other two reviewers. We tried our best to balance the view of each reviewer with the goal of streamlining the focus of the work to benefit the general readers.

2. If both *S. stipitis* and *S. cerevisiae* suffer from CCR, why not make a CCR optimized *S. cerevisiae* strain that produces norclaurine? The shikimate transport issue wouldn't then be a problem. The only answer I could find to this question was late in the paper, line 444, and in conclusion line 482.

Response: We would like to thank the reviewer for pointing out an alternative option of using a monoculture. Over the past 10 years, our group has built up a specialty for exploring various nonconventional microbial species that have unique capabilities. We emphasize pragmatic host-preselection because concurrent evaluation of a candidate host's innate capabilities and domestication potential maximizes the probability of engineering a system with the desired outcome. Although solving the CCR-relevant issues in *S. cerevisiae* could be a viable strategy for making the strain produce (S)-norcoclaurine, it will not offer the opportunity for us to elucidate the regulation mechanisms of CCR in an interesting but understudied host (i.e., *S. stipitis*) that is natively equipped with a highly efficient xylose assimilation capability.

Here is the history of this project. In our previous study(15), we have shown that transformation of a plasmid carrying the shikimate pathway into *S. stipitis* yielded 7-fold higher level of shikimate than the *S. cerevisiae* counterpart. This was presumably due to the relatively high flux in the pentose phosphate pathway (PPP) enabled by the xylose assimilation capability of *S. stipitis*. Our original thought was that *S. stipitis* might be a highly promising host for producing aromatic amino acid pathway derivatives (e.g., (S)-norcoclaurine). However, we soon realized that significant technology hurdles exist for heterologous expression of the downstream genes, many of which are sourced from plants. Decades of research efforts have been made in *S. cerevisiae* to tackle some of the hurdles but transferring those strategies to a relatively new host, still suffers from the lack of enough plasmid options, especially considering that natural product biosynthetic pathways are usually long, which would aggravate the plasmid instability issue. Also, at the beginning stage of the project, integration of a long pathway into the genome of *S. stipitis* was very challenging due to the dominance of the inherent nonhomologous end joining mechanism that prevents precise genome editing.

At the time, we proposed this plug-and-play system, i.e., in principle, if we could find an appropriate connecting molecule, we can leverage the specialty of both *S. stipitis* (high PPP flux) and *S. cerevisiae* (knowledge for heterologous expression of the downstream challenging genes) for synthesizing various aromatic amino acid pathway derivatives.

3. Line 162 – ameliorate the availability of E4P. I don't understand this, do the authors mean alleviate an E4P bottleneck? Increase E4P pools? I'm not sure.

Response: Thanks for the suggestion. We have rephrased the sentence to “As xylose is assimilated into the central metabolism through PPP, improved simultaneous utilization of glucose and xylose would presumably increase the E4P supply, leading to a higher flux towards the upstream module of the AAA biosynthesis pathway (e.g., the segment for shikimate synthesis.” (Line 108-109)

4. Line 165 – please cite a review or reviews that highlight the ubiquity of this problem.

Response: We have cited two reviews there.

“However, persistent carbon catabolite repression (CCR) severely prohibits *S. stipitis* from efficiently assimilating xylose in the presence of glucose, a ubiquitous cellular phenomenon observed in bacteria, yeasts, and other fungi(16, 17).” (Line 112-113)

5. Line 173 – Why not a hydrolysate? Yeast fermentation media are pretty complex mixtures, I’m not an expert on purification but it doesn’t seem like a mixture of glucose, xylose, lignin derivatives would be that troublesome unless there were huge amounts of furfurals and aromatics. It is plant material after all and plants are often sourced for nutraceuticals and pharmaceuticals. It seems the authors are responding to a past criticism here but I’m not sure they need to eliminate this possibility so strongly, although a purified G-X mixture is obviously preferable to a complex hydrolysate.

Response: Thanks for the suggestion. To avoid confusion, we have removed the corresponding sentence from the manuscript – “Raw biomass hydrolysates should not be considered as carbon sources, because AAA derivatives are exploited as nutraceuticals and pharmaceuticals, for which downstream purification accounts for a significant portion of the overall production cost, especially when the titer is low.” (Line 117-120)

6. Line 193 – other bottlenecks exist. This seems like a leap of a claim. Is this only because alleviating CCR on the PPP didn’t completely remove the inhibition?

Response: Yes, the reviewer’s understanding was correct. Refactoring the xylose utilization pathway only partially alleviates CCR inhibition (Figure 2a). “Other bottlenecks, such as efficient xylose import and global regulation(17), exist and need to be addressed to enable a more substantial mitigation of CCR.” (Line 139-140)

7. Line 227 – serving as repositories. I would revise this to “reservoirs” or “sources”.

Response: Thanks for the suggestions. We have revised “repositories” to “reservoirs”. (Line 174)

8. The first few lines of the Discussion are confusing as they fluidly transition from *S. cerevisiae* to *S. stipitis* without a clear difference.

Response: We would like to thank the reviewer for helping us streamline the Discussion section. We have removed the relevant sentences considering they were somewhat redundant with the texts in the Results section. We also incorporated a few more paragraphs there to respond to the requests of the other two reviewers.

9. Line 408 and 409 – xylose transport has been known as a bottleneck for at least twenty years, and Kotter, Zhao, Alper, de Waal, and Boles, among others, have done a lot of work (uncited here save for Zhao I think) to engineer transport. I think it is a little bit of an overstatement to say that it “emerged as a bottleneck”.

Response: We have removed the sentence.

10. Line 412 – I think the authors need to be more specific about what their study proves about transport and metabolic engineering because, to my previous comment, I think this has been proved many times in the literature.

Response: Following the suggestion, we have removed the corresponding sentence.

11. Line 414 – SA is used instead of shikimate as an unexplained abbreviation

Response: We have removed the abbreviation and stuck to the use of shikimate.

12. Line 422 – Candida albicans

Response: Please see the response below.

13. Line 422 – this is the first mention of Ca and Dh transporters being investigated – shouldn't all transporters be mentioned in the results?

Response: To keep the narrative of the manuscript more focused, we decided to remove this part from the manuscript. The main findings and the conclusions in our study remain unaltered.

14. Line 462 – “all that ever reported” - please revise this

Response: We have revised it to “than all previously reported yields” in the revised manuscript (Line 455-456).

15. Line 469 – I'm not sure how the paper leads to this statement. The statement seems completely separate from the point of the paper, which is transport and consortium engineering.

Response: Thanks for the suggestion. We have removed the sentence from the manuscript.

Reviewer #3 (Remarks to the Author):

The manuscript reports a consortium platform consisting of two different yeast strains to produce a target molecule. The yeast consortium demonstrated effective partitioning of metabolic pathways into different yeast strains to produce an aromatic amino acid derivative. In order to implement the yeast consortium, the authors identified a putative xylose-specific transporter for *Schefferomyces stipitis* and quinate permeases for *Saccharomyces cerevisiae* to enable enhanced production of (S)-norcoclaurine from a mixture of glucose and xylose. The presented results in the manuscript are promising, but additional experiments and revisions of the manuscript might be necessary.

Specific comments:

1. Line 188-192, Please define HCDF – high cell density fermentation?

Response: We have added the abbreviation into the manuscript — “High cell-density fermentation (HCDF, with an initial OD_{600nm} at ~ 10)” (Line 132).

2. According to S1 figure, *S. stipitis* consumed glucose efficiently even with low initial OD inoculations under YP conditions. However, in Fig 2a, it took 5 days for both control and mutant strains to consume 20-30 g/L glucose even with high-density inoculations under SC-ura conditions. These are extremely slow glucose consumption rates. Does the 20kb PMG xyl plasmid has negative effects on yeast growth?

Response: This is a good catch. The wild type *S. stipitis* grows rapidly (and thus consumes glucose quickly) in the YP (Yeast extract and peptone) medium, in contrast to the growth rate of a *ura3* auxotrophic strain transformed with a plasmid in the SC (Synthetic Complete) medium.

To answer whether the 20 kb plasmid has a negative effect on growth, we compared the growths of three *S. stipitis* strains (see the figure below), two carrying a plasmid of ~20 kb and one carrying an empty plasmid of 10.8 kb. The growth rates of the three strains were very similar to each other in the SC medium.

Figure. Cell growth profiles of *S. stipitis* variants harboring different sizes of plasmids in SC-ura medium containing 70 g/L glucose and 40 g/L xylose. Initial cell density was controlled at ~0.2. Ss-CEN5 harbors a 10.8-kb plasmid, Ss-SA7.3 harbors a 19.9-kb plasmid, and Ss-xyl harbors a 20.6-kb plasmid. Samples were collected every 24 h. Error bars represent the standard deviations of biological triplicates.

3. Line 222-223, the expression of SsXUT1, 2, and 4 indeed enhanced co-utilization of glucose and xylose. If Fig2A and B are compared, it seems that the expression of those transporters improved not only xylose uptake but also glucose consumption. Are they xylose-specific transporters? The statement that SsXUT1,2 and 4 are xylose-specific transporters without glucose inhibition might not correct. Further experimental validation such as glucose/xylose uptake kinetics with and without glucose will need to be conducted. Those transporters still might exhibit glucose transport activities.

Response: Please see the response to comment #5 below.

4. For Fig 2B, this reviewer strongly recommends providing the sugar consumption and growth profiles with the best transporter candidates. It will visualize the effects of co-consumption.

Response: Please see the response to comment #5 below.

5. Previous studies on transporter engineering by Young et al. and Farwick et al. have identified critical amino acid residues responsible for co-transport or/and xylose-specific transport. Have the authors identified any of those critical residues in Ss or SpXUTs? The comparison result can be discussed in the Discussion.

Response: We appreciate the rigorous comments here. Since all these three questions are related, we would like to combine our responses.

Firstly, we agree that the current data is not sufficient to claim the transporters are specific to xylose. Therefore, we removed “free from glucose inhibition” and revised the original sentence as below.

“The activation at the transcriptional level through promoter swapping confirms that SsXut1, SsXut2, and SsXut4 are all functional xylose transporter free from glucose inhibition.” Response to comment #3, Line 168)

Secondly, we have added Figure S3 into the manuscript, showing the sugar consumption and growth profiles of top transporter candidates. Consistent with the results shown in Figure 2, SsXut1, SsXut2, and SsXut4 all demonstrated pronouncedly faster consumption of xylose. “It was interesting to observe that SsXut4 also led to a faster glucose consumption, suggesting its potential co-transport activity toward both glucose and xylose.” (Response to comment #3 and #4)

Figure S3. The sugar utilization and cell growth profiles of the strain Ss-xyl expressing xylose transporters (a) SsXut1, (b) SsXut2, (c) SsXut4, (d) SpXut1, or (e) no transporter in SC-ura plus 70 g/L glucose and 40 g/L xylose. Initial cell density was at ~10. The Ss-xyl expressing no transporter was used as a negative control strain. Ss, *Scheffersomyces stipitis*; Sp, *Spathaspora passalidarum*. Ss-xyl carries the engineered xylose assimilation pathway with the promoters of the five identified genes swapped with constitutive ones. Error bars represent the standard deviations of biological triplicates. It was interesting to observe that SsXut4 also led to a faster glucose consumption, suggesting its potential co-transport activity toward both glucose and xylose.

Lastly, we performed protein sequence alignments and added the following texts into the Discussion section regarding the critical amino acid residues previously identified to be responsible for co-transport and/or xylose-specific transport. The reason for us not being able to perform the sugar uptake kinetic experiments was also explained in the second half of the paragraph below (Response to comment #5).

(Line 416-441) A key discovery in this study is the identification of novel transporters. For xylose transporter, Young et al. previously identified a conserved motif (G-G/F-XXX-G) enriched among multiple heterologously expressed monosaccharide transporters that conferred faster growth rates on xylose than on glucose(18). Farwick et al. studied the native hexose

transporter GAL2 in *S. cerevisiae* and found that a single point mutation N376F completely abolished glucose transport activity while still enabled effective xylose translocation(19). To understand if these critical residues exist in the transporters in the current study, the amino acid sequences of the best transporter, SpXUT1, and all seven SsXUT transporters were aligned to GXS1 from *Candida intermedia* (named CiGXS1) (18) and ScGAL2(19) (Figure S13). Interestingly, four out of the eight transporters were found to harbor the G-G/F-XXX-G motif, supporting that the conserved motif is highly enriched in the transporters that are functional for xylose. Moreover, among the eight, SpXUT1, SsXUT1, SsXUT2, and SsXUT4 contain methionine, methionine, phenylalanine, and tyrosine at the counterparts of the ScGAL2_{N376} position. Based on Farwick's study, ScGAL2_{N376Y} was demonstrated to be free of glucose inhibition although the xylose transport activity was also negatively impacted whereas ScGAL2_{N376F} enabled a strong and glucose-resistant transport of xylose. It is worth pointing out that both Young and Farwick's studies were based on *S. cerevisiae* EB.YVW4000, a widely used platform strain with at least 20 endogenous hexose transporters deleted for characterization of novel sugar transporters(20). Such a parental strain is not easily available for *S. stipitis* because sequentially knocking out more than 20 native transporters in combination of selection marker recycling is still challenging due to the dominance of the inherent nonhomologous end joining mechanism that prevents precise genome editing. Therefore, in the current yeast consortium study although SpXUT1, SsXUT1, SsXUT2, and SsXUT4 can effectively transport xylose in the presence of glucose, we cannot exclude their activities towards glucose. Additional validation such as sugar uptake kinetics experiments in a similar strain background to *S. cerevisiae* EB.YVW4000 will be needed for detailed transporter characterization studies in the future.

CiGXS1	MGLE-----DNRMVKRFV-----NV	15
ScGAL2	MAVEENNMPVVSQQPQAGEDVISSLSKDSHLSAQSQKYSNDELKAGESGSEGSQSVPIEI	60
SpXUT1	-----MHGG-----SDGNDVQ-AIIAQKRLEIAGKPGIA---	28
SsXUT1	-----MHGG-----GDGNDIT-EIIAARRLQIAGKSGVA---	28
SsXUT2	-----MKYFQI---	6
SsXUT3	-----MREVGILDVAHGNNVT---	16
SsXUT4	-----	0
SsXUT5	-----MT-----ERSIGPLI	10
SsXUT6	-----MSSV-----EKSASETASYTSQVSASGSAKTNSYL---	29
SsXUT7	-----	0
CiGXS1	GEKKAGSTAMAIIVGLFAASGGVLPFYDGTGTSISGVMTMDYVYLARYPSN----KHSPTADE	71
ScGAL2	PKKPMSEYVTVSLLCLCVAFGGFMPGWDGTGTSISGFVVQTDFLRRFGMKHKDGTHTYLSNVR	120
SpXUT1	---GLIANRKSFLIAVFASLGGLVYGYNQGMFGQISGMTSFSAAAGVG---KIQDNPTL	81
SsXUT1	---GLVANSRSFP IAVFASLGGLVYGYNQGMFGQISGMTSFSKAIGVE---KIQDNPTL	81
SsXUT2	---WKSQKQVSYAVTFTCELAFILPGIEQGIIGNLINNDFLNTFQNP-----TGSY	55
SsXUT3	---IMMKDPVVFLVILFASLGGLLPFYDQGVISGIVTMSFPAKF--P----RIFMDADY	67
SsXUT4	-----MGSLLTNEYFKDYVHNP-----TPVE	21
SsXUT5	PR---NKHLFYGSVLLMSIVHPTIMGYDSMMVGSILNLDAYVNYFHLTAATTGLNNTAAVW	67
SsXUT6	---GLRGHKLNFVSCFAGVGFLLPFYDQGVMSGLLTLPSFENTFPAM----KASNATL	82
SsXUT7	-----	0
CiGXS1	SSLIVSILSVGTFPGALCAPFLNDTLGRRWCLLSALIVFNIGAILQVIST--AIPLLCA	129
ScGAL2	TGLLIVAFINIGCAFGGIILSRGDMYGRKGLSIV-VSVYIVGIIQIASI-NKQWYQYFI	178
SpXUT1	QGLLTSILELGAWVGLMNGYVADRVRGRWSVMFG-VAWFILGVIIQACTHGANYSFILG	140
SsXUT1	QGLLTSILELGAWVGLMNGYIADRLGRKRSVVVG-VFFPFIGVIVQAVARGGNYDYLIG	140
SsXUT2	LGIIIVSIYTLGCFPGCVNMFPIGDRMGRRSKIASS-MTVITIGVALQCSSP--SVEQLMI	112
SsXUT3	KGWVFTFLLCANFGSIINTPIVDRFGRRDSITIS-CVIFVIGSAPQC--AGINTSMLPG	124
SsXUT4	VGMTIAALLEIGALFSSPIAGRVGDIVGRRRTIRYG-SPIFVVGGLVQATSV--NIVNLSP	78
SsXUT5	LGQVIATLT-----VISYFNDRKFRSSVCIS-IAISLVGVALQSAAQ--NIEMFII	116
SsXUT6	QGAVALYEIGCMSSSLATIYLGDRLGRKIMFIG-CVIVCIGAALQASAF--TIAHLTV	139
SsXUT7	-----MTFAV--NLVYFAV	12
CiGXS1	GRVIAGFGVGLISATIPLYQSETAPKWRGAIVSCYQWAITIGLFLASCVNKGEHMT--	187
ScGAL2	GRIISGLGVGGIAVLCPLMISEIAPKHLRGTLVSCYQLMITAGIFLGYCTNYGTRKSYS--	236
SpXUT1	GRFIVGVGILSMIVPLYNAEVAPPEIRGSLVALQQLAITFGIMISYWIYGTNYIGGT	200
SsXUT1	GRFVVGIGVGLSMVVPYLYNAEVSPPPIRGLVALQQLAITFGIMISYWIYGTNYIGGT	200
SsXUT2	GRFITGLGTGWETSTCPMYQAELEPPKVRGRVLCSEALFVGVGLIYAYWFDYALSPTS--	170
SsXUT3	GRAVAGLAVGQLFMVVPMYSELAPPVSRGGLVVIQQLSITIGIMISYWLVDYGTHTPIGGT	184
SsXUT4	GRLIAGIAIGFLTFTIIPCYQSEISPPDDRRGFYACLEFTGNIIGYASSIWDYGFSPFLD--	136
SsXUT5	GRIVIGFGISIGFVSSPILVSELAPPDKRGFILGLSPTSFLVGLSIIAAGVTYGTNRNAP--	174
SsXUT6	ARIITGLGTGFTITSTVPVYQSECSAPKRRGQLIMMEGSLIALGIAISYWIWDFGFYFLRND	199
SsXUT7	GRVLSGVGVGLSTMVVPSYQCEISPSSEERKLVCGEFTGNITGYALSVDYFCYFIQDI	72
	. * : * * : * ** * .	
CiGXS1	-----NSGSYRIPLAIQCLWGLLIGIGMIFLPETPRFW	220
ScGAL2	-----NSVQWRVPLGLCFAWSLFMIGALTLVPESPRYL	269
SpXUT1	G-----EGQSKAAWLVPICIQMPALILGSCIFLMPESPRWL	237
SsXUT1	G-----SGQSKASWLVPICIQLVPALLGVGIFPMPESPRWL	237
SsXUT2	-----GPIAWRLPLASQIVFAFVVFCTFTTIPESPRYM	203
SsXUT3	RCAPSHPYQGETFPNPNVDVPPGGCYQSDASWRIPFGVQIAPAVLLGIGMIFPFRSPRWL	244
SsXUT4	-----NDFSWRSPLYVQVVIGSMLFSGFLIVETPRWL	169
SsXUT5	-----GDWCWRIPSIIQGAPDIVAIINILPISPRWL	207
SsXUT6	G-----LHSSASWRAPIALQCVFAVLLISTVFFPESPRWL	235
SsXUT7	GDAREKPH-----SFFAHLRWLPLFIQVVIAAVLFGVGGFFIVESPRWL	116
	: * : * * :	
CiGXS1	ISKGNQEKAAESLARLRKL-PIDHPDSLEELRDITAAYEFETVYGKS---SWSQ-----	270

Figure S13. Protein sequence alignment of multiple sugar transporters. Ci, *Candida intermedia*; Sc, *Saccharomyces cerevisiae*; Ss, *Scheffersomyces stipitis*. The conserved motif G-G/F-XXX-G from CiGxs1 and the single point mutation in ScGAL2 and their counterparts in SpXut1 and SsXut were highlighted. The result was generated using Clustal Omega online (<https://www.ebi.ac.uk/Tools/msa/clustalo/>) with default parameters.

ScGAL2	CEVNVKVEDAKRSIAKSNKV-SPEDPAVQAELDLIMAGIEAEKLAGNA---SWGE-----	319
SpXUT1	MNEGNEEKCLDVLRLRGL-DRNNELIQMEFLEMKAQKIFHEHELEAT--AYPDLQDGSAS	294
SsXUT1	MNEDREDECLSVLSNLRSL-SKEDTLVQMEFLEMKAQKLFERELSAK--YFPHLQDGSAS	294
SsXUT2	FYKGEKEEAKRILSYVFGK-PGDHPDILKEWINDINDAVILETSEG-A--FSWAK-----	253
SsXUT3	LSKGRDEEAWSSSLKYLRRK-SHED-QVEREFAEIKAEVVYEDKYYEK--RFPQKT-----	295
SsXUT4	LDBNHDIEGMIVISDLYADGDVEDDDAIAEYRNKIKESVLIARV--EGGERSYQY-----	221
SsXUT5	IAKERPSEAREIISIIISDV-PIEDA--HEECEKIHAIHQTEKTAFPG--NKWK-----	255
SsXUT6	LNKGRTEEAREVFSALYDL-PADSEKISIQIEEIQAADLERQAGEG--FVLKE-----	286
SsXUT7	LDVDQDQQGFHVLALLYD-SHLDDNKPREFFMIKNSILLERETTPKERTWKH-----	169
	. . . : : :	
CiGXS1	----VFS-----HKNHQLKRLFTGVAIQAFQQLTGVNFIIFYGTFFPKRAGVNG-FTI	318
ScGAL2	----LFS-----TKTKVQFRLLMGVFVQMFQQLTGNMYFFYYGTVIPKSVGLDDSFET	368
SpXUT1	SRFKIGFLQYKSMLEHYPTFKRVAVACLIMTFQQTGWVNFILYYAPFIFASLGLSGKTS	354
SsXUT1	SNFLIGFNQYKSMITHYPTFKRVAVACLIMTFQQTGWVNFILYYAPFIFSLSGLSGNTIS	354
SsXUT2	----LFK-----PDKARTGVRVFLAYMSMFAQLLGSVNVVNYIITFVLINSVIGIEDNLA	303
SsXUT3	-GVALTLTGWDILTTKSHKRVF IGSVMPFQQF IGCNAI IYYAPTIFPQLGMNSTTTS	354
SsXUT4	----LFTK-----YTKRLSVACFSQMFQMQMNGINMVSYYAPMIFESAGVWGR-QA	266
SsXUT5	-----QMVSSKSNTRRVIILFTQAIIVTEMAGSSVGSYYFSIILTQAGVKDSNDR	304
SsXUT6	----LFT-----QGPARNLQRVALSWSQIMQITGINIIITYYAGTIFESYIGMSPFMS	336
SsXUT7	----MFKN-----YMTRVLIACALGFAQFNGINIIISYYAPMVFEEAGFNNS-KA	214
	*: : * . ** .:	
CiGXS1	SLAT---NIVNVGSTIPGILLMEVLGRRNMLMGATGMSLSQLLIVAIVGVATS-----	368
ScGAL2	SIVI---GVVNFASFSTLWTVENLGRKCLLLGAATMMACMVIYASVGVTRLYPHGKQ	425
SpXUT1	LLASGVVGI V MFLATIPAVLWVDLGRKPVLSIGALLMGMCHFVVAGILG---GLHGDF	410
SsXUT1	LLASGVVGI V MFLATIPAVLWVDLGRKPVLSIGAIIMGICHPVVAAILG---QPGGNF	410
SsXUT2	LILGGVAVICFTVGLVPTFFADRMRRLPSAVGAFGCGVCMMLISI-L-LL---SPQDHP	358
SsXUT3	LLGTLGYIVNCLSTLPAVFLIDRCGRKTLTMAGAIGTFISLVIVGAIIVG---KYGDRL	410
SsXUT4	ILMTGINSIIYIFSTIPWYLVDVSWGRKPLLSGSLMGMVPLLTACSLF-----	316
SsXUT5	LRVNVIMSSWSLVIALSGCLMFDRIGRKMQSLISLSGMIICFIVLGLVLK---EYGDGH	360
SsXUT6	RILAALNGTEYFLVSLIAFYTVLGRFRLLFWGAIAMALVMAGLTV-TV---KLA--G	389
SsXUT7	LLMTGINSIVYWFSTIPWFLVDHWGRKPIISGGLSMGICIGLIAVVIL-----	264
	:: : *:: .	
CiGXS1	ENNKSSQSVLVAFSCFIIAFAATWGPCAWVVVVELFPLRTRAKSVSLCTASNWLNWNGI	428
ScGAL2	PSSKGAGNCMIVPFCFYIFCYATTWAPVAWVITAESPPLRVKSKMALASASNWVWGLFI	485
SpXUT1	TNNMGAGAAVVFIWLFAIFPGYSWGPCAWVIVAEVFPGLRARGVSIAGSSNWLNNFAV	470
SsXUT1	VNBSGAGWVAVVFWIAFAGFGYSWGPCAWLVVAEVFPGLRARGVSIAGSSNWLNNFAV	470
SsXUT2	KLKSSGAGAVAFFVFVQLVFGSTGNCPWMSISELIPHLARAGSSLATSSNWLWNPFV	418
SsXUT3	SEFKTAGRTAIAFIFIYDVNFYSWAPIGWVLPSEIFPIGIRSNASITTSSTWMMNFI	470
SsXUT4	LNNTYTPGVVVGSVIVFNAAPFGYSWGPWPWL-SEVFPNVSRSKGAAMSTATNWLNFIV	375
SsXUT5	--SKSGSYAAVAMMFLFTGFSFTFTPLNSLYPPELFPYVLRSTGVTLFNIIFNGCWLFA	418
SsXUT6	EGNTHAGVGAALLFAFNSFFGVSWLGGSWLLPPELLSLKLRAPGAALSTASNWAPNFMV	449
SsXUT7	LDKSFTPSMVAVLVIIYNASFGYSWGPFGFLIPPEVPLAVRSKGVSISTATNWFANFVV	324
	: .: : * : : : .	
CiGXS1	AYATPYMVDEDRGNLGSNVFFIWWGGFNLACVFFAWYFIYETKGLSLEQVDELYEHVSKAW	488
ScGAL2	AFPTPITSAINFYY---GYVFMGCLVAMFFYYVFFVPETKGLSLEEIQELWEEGVLPW	541
SpXUT1	AMSTPDPVAKATYGA---YIFLGLMCFVGAAYVFFCPETKGRITLDEIDELFGDTSQVS	526
SsXUT1	AMSTPDPVAKAKFGA---YIFLGLMCFVGAAYVQFPCPETKGRITLDEIDELFGDTSQVS	526
SsXUT2	VEITPTIEKLKWK---YLIFMCCNFSVPVMPYFFFPETKNTLEAIDDLFS-----	468
SsXUT3	GLVTPHMLETMKWT---YIFFAAFAIIAFFFTWLIIPETKGVPLEEMDAVFGDTAALQ	526
SsXUT4	GEMTPILLDTITWRT---YLIPATSCVLSFPVAVGLFPETKGLALEDMGVSFDDNSSIF	431
SsXUT5	SFILPIAMNGIGWKF---YIINACYDVIPLIIMFCWIETKGINLDTISEVLHGRGPE	474
SsXUT6	VMITPVGFQSIGSYT---YLIFAAINLLMAPVIYFLYPETKGRSLEEMDIIFNQCVPWE	505
SsXUT7	GQMTPILOQLRWGT---YLFPAGSICIISVIVVIFYPETKGVLEEDMDVSVFESFY---	377
	* . . ***. * : :	
CiGXS1	KSKGFVPSKHSFREQ-----VDQQMSKTE-----AIMSEEASV-----	522
ScGAL2	KSEGWIPSSRRGN---DLEDLQHDDKPW---YKAMLE-----	574
SpXUT1	KREGEIRN-RILKE---VGLLELIGLEELSDK---SKGGDVHYQEEKAADADADSA---	576
SsXUT1	RMEKEIHE-QKLE---VGLLQLLGEENASESE---NSKADVYHVEK-----	566
SsXUT2	-----	468
SsXUT3	EKNLV-----TITSVSEDAKD---RN--SIEMSE-----	551

Figure S13 (cont.). Protein sequence alignment of multiple sugar transporters. Ci, *Candida intermedia*; Sc, *Saccharomyces cerevisiae*; Ss, *Scheffersomyces stipitis*. The conserved motif G-G/F-XXX-G from CiGxs1 and the single point mutation in ScGAL2 and their counterparts in SpXut1 and SsXut were highlighted. The result was generated using Clustal Omega online (<https://www.ebi.ac.uk/Tools/msa/clustalo/>) with default parameters.

6. Line 238-239 – Hxt11 mutant transporter has been reported to be a co-transporter of glucose and xylose in a hxt-null background with reduced kinetic properties. Given that Ss strain has native xylose transporters intact, the statement, “neither transporter conferred xylose uptake in *S. stipitis*” does not make sense.

Response: We have revised the sentence to “neither transporter improved xylose uptake in *S. stipitis*” in the manuscript (Line 188).

7. Line 241-244, while fermentation data showed enhanced xylose uptake in the presence of glucose, further sugar uptake kinetics experiment might be necessary to support the statement “xylose-specific transporter”.

Response: We agree with the reviewer’s comment that the current data is not sufficient to support the statement of xylose-specific transporters and have removed the relevant texts. The following discussion was added to clarify why sugar uptake kinetics experiments based on a *hxt*-null and *xut*-null strain background is still challenging for *S. stipitis*.

(Line 431-441) It is worth pointing out that both Young and Farwick’s studies were based on *S. cerevisiae* EB.Y.VW4000, a widely used platform strain with at least 20 endogenous hexose transporters deleted for characterization of novel sugar transporters(20). Such a parental strain is not easily available for *S. stipitis* because sequentially knocking out more than 20 native transporters in combination of selection marker recycling is still challenging due to the dominance of the inherent nonhomologous end joining mechanism that prevents precise genome editing. Therefore, in the current yeast consortium study although SpXUT1, SsXUT1, SsXUT2, and SsXUT4 can effectively transport xylose in the presence of glucose, we cannot exclude their activities towards glucose. Additional validation such as sugar uptake kinetics experiments in a similar strain background to *S. cerevisiae* EB.Y.VW4000 will be needed for detailed transporter characterization studies in the future.

8. Line 350-352 – the observed that *S. cerevisiae* overexpressing Aro1 or AroL with Aro7mut exhibited growth defects when shikimate is present. Why previously constructed strains with the addition of the upstream pathways (i.e., shikimate overproducing) did not show this toxicity? Maybe intracellular metabolomics study can be conducted to clarify this.

Response: This is an interesting point. We postulated the expression of the endogenous ARO1 is dynamically regulated as opposed to constitutive expression. We have added Figure 4a into the manuscript and discussed the possible reasons.

(Line 305-312) We postulated that overexpression of AnQut1co and shikimate kinase (encoded by AroL or Aro1) enabled shikimate import and increased the conversion of shikimate to shikimate-3-phosphate (S3P) in a reaction that uses ATP as a cofactor (Figure 4a). It is possible that the excessive drainage of the energy molecule in the presence of high-level shikimate led to poor cell propagation. Alternatively, whether the accumulation of certain downstream intracellular metabolite(s) resulting from overexpressed shikimate kinase may have inhibited cell growth remains as another possibility to be investigated in the future.

Figure 4. Screening of transporters for efficient shikimate uptake by *S. cerevisiae*. (a) The shikimate-to-tyrosine pathway.

9. In Fig 5b, the authors aimed to demonstrate the beneficial effects of sequential inoculation of the strains on S-norcochlorine production. However, after initial inoculation of the NC3 strain into SA4 grown media, S-norcochlorine concentration did not change within the next 96 hours (even decreased to some extent till 72 hr). It is quite difficult to understand Fig 5b. It would be helpful to provide a full fermentation profile showing sugar consumption, shikimate, S-norcochlorine production, and other metabolites (ethanol, glycerol, and acetate) in the best condition that yielded 4.2 mg/L.

Response: Following the suggestions, we have conducted more comprehensive experiments to optimize the condition. The co-culture of SA4/NC1 under the best condition (an initial OD_{600nm} of three for NC1 and a mixing time of 0 h) resulted in the highest (S)-norcochlorine titer (11.5 mg/L) (Figure 5c). Meanwhile, by-products (ethanol, glycerol, and acetate), remaining shikimate, cell density (OD_{600nm}), and residual glucose and xylose in the co-culture were measured during the 120-h fermentation (Figure S12). We have added the following texts into the manuscript.

(Line 398-413) Under the optimized condition of simultaneously mixing NC1 and SA4 with initial OD_{600nm} of three and 0.2-0.3, respectively, the highest (S)-norcochlorine titer of the Ss/Sc consortium (11.5 mg/L) was obtained. This was almost 110-fold and 37-fold higher than those reported in the literature for a *S. cerevisiae* monoculture(5) and a *S. stipitis* monoculture(6), respectively, supporting a strong synergy between the two species in the consortium. In addition, more than three-fold higher accumulation of dopamine was observed during the coculture. This is consistent with the previous monoculture studies in *S. cerevisiae*(5) and *S. stipitis*(6) that reported dopamine accumulation, suggesting that either low NCS expression level or insufficient 4-HPAA precursor supply limited (S)-norcochlorine formation. The native Ehrlich pathway could also play a role as it might rapidly transform 4-HPAA to the corresponding fusel acid, 4-hydroxyphenylacetic acid (4-HPAC) or alcohol tyrosol(4). It was also observed that the mixed sugars (including ~20 g/L glucose and ~20 g/L xylose) were rapidly converted to the main side product ethanol and a minor amount of glycerol via the Crabtree-positive yeast NC1, reaching the highest levels at 12 h (Figure S12). From 12 h to 48 h, ethanol and glycerol were consumed along with xylose and residual glucose to continuously support energy metabolism, biomass formation, and production of the target compound, its intermediates, and acetate as a byproduct.

Figure 5. Optimizing (S)-norcochlorine production by yeast consortia.....(c) The production time course of (S)-norcochlorine and dopamine in the consortium under the optimal fermentation condition. The initial cell density of SA4 was 0.2-0.3 and NC1 was simultaneously introduced into the culture with an OD_{600nm} of three. Samples were collected every 24 h post the starting of the co-culture. Error bars represent the standard deviations of biological quadruplicates.

Figure S12. Measurements of (a) by-products (ethanol, glycerol, and acetate) and (b) shikimate accumulation, cell density (OD_{600nm}), and residual glucose and xylose in the coculture consisting of SA4 and NC1 under the optimal co-culture condition. The strains SA4 and the NC1 were simultaneously introduced into the culture with an OD_{600nm} of 0.2-0.3, and three, respectively. Samples were collected every 24 h post the starting of the co-culture. Error bars represent the standard deviations of biological quintuplicates.

10. Liu et al. Rewiring carbon metabolism in yeast for high-level production of aromatic citation appear twice in reference and main text (41, 50).

Response: Thanks for the suggestion. We have corrected the mistake.

Reference

1. M. E. Pyne *et al.*, A yeast platform for high-level synthesis of tetrahydroisoquinoline alkaloids. *Nat Commun* **11**, 3337 (2020).
2. C. G. Conacher, R. K. Naidoo-Blossop, D. Rossouw, F. F. Bauer, Real-time monitoring of population dynamics and physical interactions in a synthetic yeast ecosystem by use of multicolour flow cytometry. *Appl Microbiol Biotechnol* **104**, 5547-5562 (2020).
3. S. H. Yalkowsky, Y. He, P. Jain, *Handbook of aqueous solubility data*. (CRC Press, Boca Raton, FL, ed. 2nd, 2010), pp. xii, 1608 p.
4. M. Cao, M. Gao, M. Suastegui, Y. Mei, Z. Shao, Building microbial factories for the production of aromatic amino acid pathway derivatives: From commodity chemicals to plant-sourced natural products. *Metab Eng*, (2019).
5. W. C. DeLoache *et al.*, An enzyme-coupled biosensor enables (S)-reticuline production in yeast from glucose. *Nat. Chem. Biol.* **11**, 465-471 (2015).
6. Y. Zhao *et al.*, Leveraging the Hermes Transposon to Accelerate the Development of Nonconventional Yeast-based Microbial Cell Factories. *ACS Synth Biol* **9**, 1736-1752 (2020).
7. X. Liu *et al.*, Convergent engineering of syntrophic Escherichia coli coculture for efficient production of glycosides. *Metab Eng* **47**, 243-253 (2018).
8. S. Freilich *et al.*, Competitive and cooperative metabolic interactions in bacterial communities. *Nat Commun* **2**, 589 (2011).
9. W. R. Harcombe *et al.*, Metabolic resource allocation in individual microbes determines ecosystem interactions and spatial dynamics. *Cell Rep* **7**, 1104-1115 (2014).
10. A. Kerner, J. Park, A. Williams, X. N. Lin, A programmable Escherichia coli consortium via tunable symbiosis. *PLoS One* **7**, e34032 (2012).
11. X. Li *et al.*, Design of stable and self-regulated microbial consortia for chemical synthesis. *Nat Commun* **13**, 1554 (2022).
12. R. Wang, S. Zhao, Z. Wang, M. A. Koffas, Recent advances in modular co-culture engineering for synthesis of natural products. *Curr Opin Biotechnol* **62**, 65-71 (2020).
13. M. R. Antoniewicz, A guide to deciphering microbial interactions and metabolic fluxes in microbiome communities. *Curr. Opin. Biotechnol.* **64**, 230-237 (2020).
14. M. Gao, D. Ploessl, Z. Shao, Enhancing the co-utilization of biomass-derived mixed sugars by yeasts. *Front. Microbiol.* **9**, (2019).
15. M. R. Gao *et al.*, Innovating a Nonconventional Yeast Platform for Producing Shikimate as the Building Block of High-Value Aromatics. *ACS Synth Biol* **6**, 29-38 (2017).
16. T. Franzino *et al.*, Implications of carbon catabolite repression for plant-microbe interactions. *Plant Communications* **3**, 100272 (2022).
17. K. J. Fox, K. L. J. Prather, Carbon catabolite repression relaxation in Escherichia coli: global and sugar-specific methods for glucose and secondary sugar co-utilization. *Current Opinion in Chemical Engineering* **30**, 9-16 (2020).
18. E. M. Young, A. Tong, H. Bui, C. Spofford, H. S. Alper, Rewiring yeast sugar transporter preference through modifying a conserved protein motif. *Proc. Natl. Acad. Sci. U. S. A.* **111**, 131-136 (2014).

19. A. Farwick, S. Bruder, V. Schadeweg, M. Oreb, E. Boles, Engineering of yeast hexose transporters to transport D-xylose without inhibition by D-glucose. *Proc Natl Acad Sci U S A* **111**, 5159-5164 (2014).
20. R. Wieczorke *et al.*, Concurrent knock-out of at least 20 transporter genes is required to block uptake of hexoses in *Saccharomyces cerevisiae*. *FEBS Lett* **464**, 123-128 (1999).

Reviewers' Comments:

Reviewer #1:

Remarks to the Author:

The revised manuscript has addressed most of the previous review comments. New experimental results and discussion were added, which improves the quality of this work for publication. The following issues need to be further addressed.

For the pathway metabolite accumulation profiles in Fig. 5, it is important to add the shikimate concentration change with time, which can reflect the dynamic change of the upstream and downstream consortium strains' shikimate generation and consumption strengths. The comparison of the shikimate concentration with (S)-norcochlorine and dopamine concentrations can also indicate whether the bottleneck step(s) is in the upstream or downstream module.

The previous question about the contribution of the overexpressed genes to the xylose utilization is not well answered. It would be helpful to discuss which recruited gene(s) is more effective for the xylose pathway refactoring, as it would provide guidance for future studies in this area. The result in Fig. S4b seems to suggest that the exclusion of the *xyl1* gene impacted the xylose utilization the most whereas the exclusion of the *SpXut1* gene impacted it the least. The growth difference between these two cases is significant. However, the xylose uptake profiles in Fig. S4a do not show a similar trend. On the other hand, the use of different xylose-related genes also changed the glucose uptake rate to some degree. Please provide more discussion about this, although some of the explanation in response to reviewer 3's question on *SsXut1-7* transporters is also relevant to the *SpXut1* gene here.

Reviewer #2:

Remarks to the Author:

The authors have responded to my original comments and I support publication of this manuscript in Nature Communications.

Eric Young

Reviewer #3:

Remarks to the Author:

The authors conducted additional experiments and revised the manuscript appropriately based on review comments.

Reviewer #1 (Remarks to the Author):

The revised manuscript has addressed most of the previous review comments. New experimental results and discussion were added, which improves the quality of this work for publication. The following issues need to be further addressed.

1. For the pathway metabolite accumulation profiles in Fig. 5, it is important to add the shikimate concentration change with time, which can reflect the dynamic change of the upstream and downstream consortium strains' shikimate generation and consumption strengths. The comparison of the shikimate concentration with (S)-norcochlorine and dopamine concentrations can also indicate whether the bottleneck step(s) is in the upstream or downstream module.

Response: Thanks for the suggestion. We have moved the shikimate concentration profile from Figure S12 to Figure 5.

Figure 5. Optimizing (S)-norcochlorine production by yeast consortia. (a) (S)-norcochlorine titers of the consortia inoculated with different initial *S. cerevisiae* NC1 cell densities (OD_{600nm}). The initial cell density of *S. stipitis* SA4 was kept at 0.2-0.3 and the two strains were simultaneously introduced into the culture. Samples were collected after 96 h of fermentation. (b) (S)-norcochlorine titers of the consortia with a sequential inoculation strategy. *S. stipitis* SA4 was first grown in fermentation medium (2x SC medium containing 30 g/L glucose, 30 g/L xylose, and 5 g/L L-ascorbic acid). After 0 h, 3 h, 6 h, 9 h, 12 h, 24 h, 36 h and 48 h, *S. cerevisiae* NC1 was introduced with an OD_{600nm} of three into the corresponding *S. stipitis* SA4 culture. Samples were collected after 96 h of fermentation. (c) The production time course of (S)-norcochlorine, dopamine, and shikimate in the consortium under the optimal fermentation condition. The initial cell density of SA4 was 0.2-0.3 and NC1 was simultaneously introduced into the culture with an OD_{600nm} of three. Samples were collected every 24 h post the starting of the co-culture. Data are presented as mean \pm S.D., $n = 3$ per group for (a) and (b) and $n = 5$ per group for (c). Source data are provided as a Source Data file.

2. The previous question about the contribution of the overexpressed genes to the xylose utilization is not well answered. It would be helpful to discuss which recruited gene(s) is more effective for the xylose pathway refactoring, as it would provide guidance for future studies in this area. The result in Fig. S4b seems to suggest that the exclusion of the *xyl1* gene impacted the xylose utilization the most whereas the exclusion of the *SpXut1* gene impacted it the least. The growth difference between these two cases is significant. However, the xylose uptake profiles in Fig. S4a do not show a similar trend. On the other hand, the use of different xylose-related genes also changed the glucose uptake rate to some degree. Please provide more discussion about this, although some of the explanation in response to reviewer 3's question on *SsXut1-7* transporters is also relevant to the *SpXut1* gene here.

Response: To present the data more clearly, we have decided to show the consumption of glucose and xylose separately. We greatly appreciate the reviewer's meticulous examination, and to provide greater precision, it is important to note that the exclusion of the *Xyl1* gene impacted the OVERAL GROWTH the most whereas the exclusion of the *SpXut1* gene impacted it the least. The adverse impact of removing *Xyl1* was mediated by the reduced utilization of both glucose and xylose. A similar trend was observed when excluding *Xyl2*, *Xyl3*, or *Tkt1*. It is logical to infer that the exclusion of *SpXut1* primarily affected xylose utilization, resulting in a less pronounced impact on the overall growth compared to the other exclusions.

We have incorporated these revisions into the manuscript.

"In contrast to the control strain harboring the complete set of six genes, removal of any gene from the combination resulted in a drastically reduced efficiency of mixed-sugar utilization, with xylose assimilation being particularly affected (Figure S4). Among them, the exclusion of the *Xyl1* gene had a profound effect on overall growth, while the exclusion of the *SpXut1* gene had the least impact. The removal of *Xyl1* significantly hindered growth due to the reduced utilization of both glucose and xylose. A comparable pattern was observed when excluding *Xyl2*, *Xyl3*, or *Tkt1*. This might be due to the impacts of an attenuated cofactor balance and/or the reoccurrence of CCR on central metabolism (1). The exclusion of *SpXut1* primarily resulted in a reduction in xylose utilization, leading to a comparatively less pronounced impact on overall growth compared to the other exclusions. Further investigation using techniques like ¹³C-metabolic flux analysis is necessary to unravel the intricate relationship between sugar utilization and biomass production in these six variants. Nevertheless, the expression of the heterologous xylose transporter, *SpXut1*, in combination with promoter swapping of the genes involved in xylose conversion alleviated CCR, enabled simultaneous assimilation of 28 g/L glucose and 12 g/L xylose within 36 h in *S. stipitis* (Figure 3a).

Supplementary Figure 4. The profiles of glucose consumption (a), xylose consumption (b), and cell growth (c) of the *S. stipitis* strains cultured in SC-ura medium containing 70 g/L glucose and 40 g/L xylose. The derived variants were created by removing one gene at a time from the control strain Ss-xyl-SpXUT1 (i.e., the refactored xylose conversion pathway plus SpXut1 transporter). HCDF was conducted with an initial cell density of ~10. Samples were collected every 24 h. Data are presented as mean \pm S.D., $n = 3$ per group. Source data are provided as a Source Data file.

Reference

1. M. Gao, D. Ploessl, Z. Shao, Enhancing the co-utilization of biomass-derived mixed sugars by yeasts. *Front. Microbiol.* **9**, (2019).